# Multi-Objective Causal Bayesian Optimization

**Shriya Bhatija** [1 2]  **Paul-David Zuercher** [2 3]  **Jakob Thumm** [1]  **Thomas Bohné** [2]

## Abstract

In decision-making problems, the outcome of an intervention often depends on the causal relationships between system components and is highly costly to evaluate. In such settings, causal Bayesian optimization (CBO) exploits the causal relationships between the system variables and sequentially performs interventions to approach the optimum with minimal data. Extending CBO to the multi-outcome setting, we propose *multi-objective causal Bayesian optimization* (MO-CBO), a paradigm for identifying Pareto-optimal interventions within a known multi-target causal graph. Our methodology first reduces the search space by discarding sub-optimal interventions based on the structure of the given causal graph. We further show that any MO-CBO problem can be decomposed into several traditional multi-objective optimization tasks. Our proposed MO-CBO algorithm is designed to identify Pareto-optimal interventions by iteratively exploring these underlying tasks, guided by relative hypervolume improvement. Experiments on synthetic and real-world causal graphs demonstrate the superiority of our approach over non-causal multi-objective Bayesian optimization in settings where causal information is available.

## 1. Introduction

Decision-making problems arise in various domains, such as healthcare, manufacturing, and public policy, and involve manipulating variables to obtain an outcome of interest. In many such domains, interventions are inherently costly, and practical applications are subject to budgetary constraints. Moreover, these systems are often governed

---

[1]Department of Computer Engineering, Technical University of Munich, Munich, Germany [2]Department of Engineering, University of Cambridge, Cambridge, United Kingdom [3]The Alan Turing Institute, London, United Kingdom. Correspondence to: Shriya Bhatija <shriya.bhatija@tum.de>.

*Proceedings of the 42nd International Conference on Machine Learning*, Vancouver, Canada. PMLR 267, 2025. Copyright 2025 by the author(s).

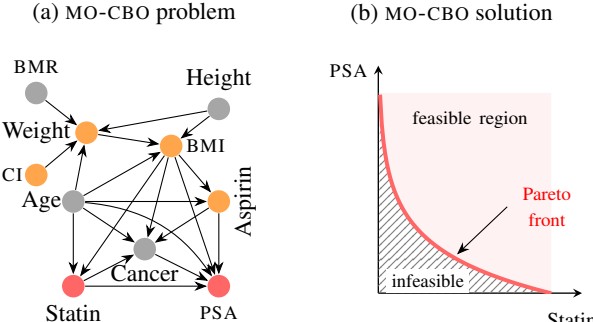

(a) MO-CBO problem     (b) MO-CBO solution

*Figure 1.* MO-CBO problem in healthcare. (a) Causal graph where red, orange, and grey nodes depict outcome, manipulative, and non-manipulative variables, respectively. (b) The solution consists of interventions that yield optimal trade-offs between the targets.

by causal mechanisms, which can be exploited to approach optimal outcomes in a targeted and cost-efficient manner. A well-established strategy for optimizing such expensive-to-evaluate black-box functions is Bayesian optimization (Shahriari et al., 2016), but it cannot leverage the causal structure between its input variables. To this end, causal Bayesian optimization (CBO) (Aglietti et al., 2020) was introduced to generalize Bayesian optimization to settings where causal information is available. While existing CBO variants focus on optimizing a single objective, real-world systems often require the simultaneous optimization of multiple outcome variables. Here, the aim is to establish optimal trade-offs between these variables instead of identifying the global optimum of a single objective. As an example, consider the graph in Figure 1 (a), which depicts the causal relationships between prostate-specific antigen (PSA) and its risk factors (Ferro et al., 2015). For patients sensitive to Statin medications, the aim is to manipulate these risk factors to simultaneously minimize both the required Statin intake and PSA levels. Figure 1 (b) shows the Pareto front, i.e., the optimal trade-offs between Statin and PSA.

We propose *multi-objective causal Bayesian optimization* (MO-CBO) to generalize CBO to problems with multiple outcome variables. Figure 2 gives a high-level overview of our proposed methodology. Our key contributions are:

1. We formally define MO-CBO as a new class of optimization problems.

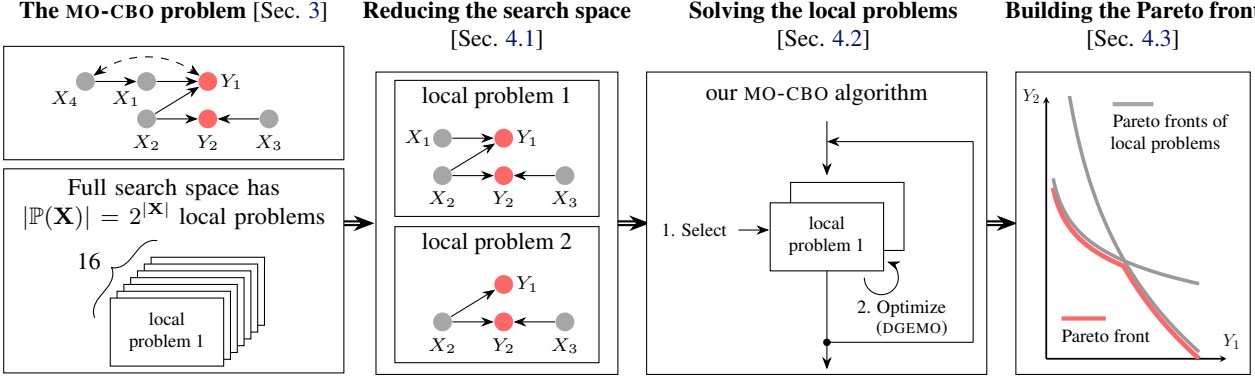

Figure 2. Overview of our MO-CBO methodology.

2. We present a mathematical framework to reduce the search space of MO-CBO problems based on the topology of the causal graph. It allows us to discard sub-optimal interventions and focus exploration on possibly-optimal strategies.

3. We propose an algorithm for the parallel exploration of these possibly-optimal intervention strategies, guided by a custom acquisition function.

4. We experimentally demonstrate on both synthetic and real-world MO-CBO problems that our method can surpass traditional multi-objective Bayesian optimization in scenarios with known causal structures, achieving more cost-effective, diverse, and accurate solutions.

To our knowledge, no other multi-objective optimization method exists in the literature that can consider the causal structure. We prove that MO-CBO's reduced search space retains all solutions achievable by traditional multi-objective optimization, while in some cases containing superior, otherwise unattainable solutions. The empirical results confirm that our MO-CBO algorithm consistently matches, and in some scenarios exceeds, the performance of standard baselines.

### 1.1. Related Work

We combine multi-objective Bayesian optimization (MOBO) with techniques from causal inference to achieve MO-CBO. Our method lies within the field of causal decision-making, seeking to leverage known causal structures to enable causally-informed decisions.

**MOBO** Multi-objective Bayesian optimization aims to efficiently optimize multiple, often conflicting, objective functions simultaneously. The existing algorithms can be roughly categorized by their selection strategy: Single-point methods select and evaluate one candidate solution at each

iteration, while batch methods select multiple solutions simultaneously for parallel evaluation. One of the most prominent single-point algorithms is ParEGO (Knowles, 2006), which randomly scalarizes the multi-objective problem into a single-objective one and chooses a sample that maximizes the expected improvement. As for batch methods, MOEA/D-EGO (Zhang et al., 2010) builds on ParEGO to incorporate multiple scalarization weights and perform batch evaluation through MOEA/D (Zhang & Li, 2007). Moreover, TSEMO (Bradford et al., 2018) adopts Thompson sampling on the Gaussian process posterior as an acquisition function, optimizes multiple objectives with NSGA-II (Deb et al., 2002), and selects the next batch of samples by maximizing the hypervolume improvement. Recently, qNEHVI (Daulton et al., 2021) was proposed as a robust method that scales to highly parallel evaluations of noisy objectives. DGEMO (Konakovic Lukovic et al., 2020) is most relevant for our work, and employs a novel batch selection strategy maintaining sample diversity in the input space. Specifically, it partitions the input space into so-called *diversity regions* to guide the selection of diverse points in each batch. We use DGEMO in our MO-CBO algorithm to explore the possibly-optimal intervention strategies.

**Causal Decision-Making** Within this field, there is a line of work focusing on multi-armed bandit problems and reinforcement learning settings. Here, actions or arms correspond to interventions on an arbitrary causal graph with existing links between the agent's decisions and the received rewards. Lee & Bareinboim (2018) identify a set of possibly-optimal arms that an agent should explore to maximize its expected reward in a multi-armed bandit problem. Moreover, Lee & Bareinboim (2019b) extend their previous work to scenarios with non-manipulative variables. Collectively, their findings represent the single-objective counterpart of our search space reduction.

Causal decision-making also encompasses a growing body of research specifically focused on advancing CBO (Aglietti

et al., 2020). These advancements include extensions such as constrained CBO (Aglietti et al., 2023), time-dynamic CBO (Aglietti et al., 2021), and various other variants (Branchini et al., 2023; Gultchin et al., 2023; Sussex et al., 2023; 2024; Ren & Qian, 2024; Zeitler & Astudillo, 2024). However, these methods are designed to optimize a single target variable, rendering them infeasible for applications with multiple objectives.

## 2. Preliminaries

In this paper, random variables and their realizations are denoted in the upper and lower case Latin letters, respectively. Sets and vectors are written in bold. For a set $\mathbf{X}$, its power set is denoted as $\mathbb{P}(\mathbf{X})$.

### 2.1. MOBO Notation

MOBO simultaneously minimizes (or maximizes) a set of black-box objectives $f_1, \ldots, f_m : \mathbf{X} \to \mathbb{R}$, where $\mathbf{X}$ is an arbitrary input space. It is designed to rely only on a small number of function evaluations. Due to potential conflicts between objectives, MOBO aims to find trade-off solutions, known as Pareto optima (Miettinen, 1999):

**Definition 2.1** (Pareto optimality). A point $x \in \mathbf{X}$ is called *Pareto-optimal* if there is no other $x' \in \mathbf{X}$ such that $f_i(x) \geq f_i(x')$ for all $1 \leq i \leq m$ and $f_i(x) > f_i(x')$ for at least one $1 \leq i \leq m$. The set of Pareto-optimal points in $\mathbf{X}$ is called *Pareto set*, denoted $\mathcal{P}_s$. The *Pareto front* is the image of the Pareto set under the objective functions, given by $\mathcal{P}_f = \{\mathbf{f}(x) = (f_1(x), \ldots, f_m(x)) \mid x \in \mathcal{P}_s\}$.

At each iteration of a MOBO algorithm, prior data is used to fit a *surrogate model* of the objectives, for which Gaussian processes (Rasmussen, 2004) are predominantly used. Based on the surrogates, an approximation $\tilde{\mathcal{P}}_f$ of the Pareto front is computed. To select which point, or batch of points, to evaluate next, an *acquisition function* is used to assess the utility of those evaluations. The most commonly used acquisition function in MOBO is based on the hypervolume indicator $\mathcal{H}$ (Zitzler & Thiele, 1999). The larger the hypervolume, the better $\tilde{\mathcal{P}}_f$ approximates the true Pareto front. The hypervolume improvement determines how much the hypervolume would increase if a batch of samples $\mathbf{B} \subseteq \mathbf{X}$ was added to the current approximation, and is given by

$$\text{HVI}(\mathbf{f}(\mathbf{B}), \tilde{\mathcal{P}}_f) = \mathcal{H}(\tilde{\mathcal{P}}_f \cup \mathbf{f}(\mathbf{B})) - \mathcal{H}(\tilde{\mathcal{P}}_f). \quad (1)$$

Since DGEMO is the backbone of our MO-CBO algorithm, we briefly describe its batch selection strategy. It considers hypervolume improvement as well as sample diversity in the input space. To this end, the so called diversity regions $\mathcal{R}_1, \ldots, \mathcal{R}_K \subseteq \mathbf{X}$ are constructed by using the current Pareto front approximation to group the optimal points

based on their performance properties in the input space. Formally, a batch is chosen as follows:

$$\mathbf{B} = \underset{\mathbf{B} \subseteq \mathbf{X}, |\mathbf{B}|=B}{\arg \max} \ \text{HVI}(\mathbf{f}(\mathbf{B}), \tilde{\mathcal{P}}_f)$$
$$\text{s.t.} \ \max_{1 \leq k \leq K} \delta_k(\mathbf{B}) - \min_{1 \leq k \leq K} \delta_k(\mathbf{B}) \leq 1, \quad (2)$$

where $B$ denotes the batch size and the functions $\delta_k(\cdot)$ are defined as the number of elements from $\mathbf{B}$ that belong to $\mathcal{R}_k$. We refer to Konakovic Lukovic et al. (2020) for the complete selection algorithm.

### 2.2. Causality

**Graph Notation** A graph $\mathcal{G} = (\mathbf{V}, \mathbf{E})$ is defined by a finite vertex set $\mathbf{V}$ and an edge set $\mathbf{E} \subseteq \mathbf{V} \times \mathbf{V}$, containing ordered pairs of distinct vertices. The subgraph of $\mathcal{G}$ restricted to $\mathbf{V}' \subseteq \mathbf{V}$ is given by $\mathcal{G}[\mathbf{V}'] = (\mathbf{V}', \mathbf{E}[\mathbf{V}'])$, where $\mathbf{E}[\mathbf{V}'] = \{(i,j) \in \mathbf{E} \mid i, j \in \mathbf{V}'\}$. For $V \in \mathbf{V}$, the set of its parents, ancestors and descendants in $\mathcal{G}$ is denoted as $\text{pa}(V)_{\mathcal{G}}$, $\text{an}(V)_{\mathcal{G}}$, and $\text{de}(V)_{\mathcal{G}}$, respectively. Here, no vertex is a parent, an ancestor, or a descendant of itself. Conversely, with a capital letter, this notation is extended to include the argument in the result, i.e., $\text{Pa}(V)_{\mathcal{G}} = \text{pa}(V)_{\mathcal{G}} \cup \{V\}$. Moreover, we define these relations for sets of variables $\mathbf{V}' \subseteq \mathbf{V}$, i.e., $\text{pa}(\mathbf{V}')_{\mathcal{G}} = \bigcup_{V \in \mathbf{V}'} \text{pa}(V)_{\mathcal{G}}$ and $\text{Pa}(\mathbf{V}')_{\mathcal{G}} = \bigcup_{V \in \mathbf{V}'} \text{Pa}(V)_{\mathcal{G}}$. Equivalent conventions apply to the ancestor and descendant relationships.

**Structural Causal Models** Let $\langle \mathbf{V}, \mathbf{U}, \mathbf{F}, P(\mathbf{U}) \rangle$ be a structural causal model (SCM) (Pearl, 2000) and $\mathcal{G}$ its associated acyclic graph that encodes the underlying causal mechanisms. Specifically, $\mathbf{U}$ is a set of independent exogenous random variables distributed according to the probability distribution $P(\mathbf{U})$, $\mathbf{V}$ is a set of endogenous random variables, and $\mathbf{F} = \{f_V\}_{V \in \mathbf{V}}$ is a set of deterministic functions such that $V = f_V(\text{pa}(V)_{\mathcal{G}}, \mathbf{U}^V)$, where $\mathbf{U}^V \subseteq \mathbf{U}$ is the set of exogenous variables affecting $V \in \mathbf{V}$. The set $\mathbf{U}^V \cap \mathbf{U}^W$ consists of unobserved confounders between $V, W \in \mathbf{V}$, which are the exogenous variables influencing both $V$ and $W$. Within $\mathbf{V}$, there are three different types of variables to be distinguished: Non-manipulative variables $\mathbf{C}$ that cannot be modified, treatment variables $\mathbf{X}$ which can be set to specific values, and output variables $\mathbf{Y} = \{Y_1, \ldots, Y_m\}$ which represent the outcome of interest. We consider only real-valued SCMs, where all endogenous variables have continuous domains. For $\mathbf{X}_s \subseteq \mathbf{X}$, $\text{CC}(\mathbf{X}_s)_{\mathcal{G}}$ refers to the $c$-component of $\mathcal{G}$ (Tian & Pearl, 2002), which, in this context, is the maximal set of variables that includes $\mathbf{X}_s$ and is connected via unobserved confounders. The joint distribution of $\mathbf{V}$, which is determined by $P(\mathbf{U})$, is referred to as *observational distribution* and denoted $P(\mathbf{V})$.

**Interventions** A set $\mathbf{X}_s \in \mathbb{P}(\mathbf{X})$ is called an intervention set. The interventional domain of an intervention set is

given as $\mathcal{D}(\mathbf{X}_s) = \times_{X \in \mathbf{X}_s} \mathcal{D}(X)$ and describes the feasible values of $\mathbf{X}_s$. An *intervention* on $\mathbf{X}_s$ involves replacing the structural equations $f_X$ with a constant intervention value $x$, for all $X \in \mathbf{X}_s$. This action is denoted with the do-operator $\mathrm{do}(\mathbf{X}_s = \mathbf{x}_s)$, where the vector of intervention values is $\mathbf{x}_s \in \mathcal{D}(\mathbf{X}_s)$. The graph $\mathcal{G}_{\overline{\mathbf{X}}_s}$ represents this intervention and is obtained by removing the incoming edges into $\mathbf{X}_s$. The observational distribution of $\mathcal{G}_{\overline{\mathbf{X}}_s}$ is denoted as $P(\mathbf{V}|\mathrm{do}(\mathbf{X}_s = \mathbf{x}_s))$ and called *interventional distribution*. For $\mathbf{X}_s = \varnothing$, no intervention is performed and the observational and interventional distributions coincide. The tuple $(\mathbf{X}_s, \mathbf{x}_s)$ is referred to as an *intervention set-value* pair. Given two sets $\mathbf{X}_s, \mathbf{X}'_s \subseteq \mathbf{X}$ and $\mathbf{x}_s \in \mathcal{D}(\mathbf{X}_s)$, we write by $\mathbf{x}_s[\mathbf{X}'_s]$ the values of $\mathbf{x}_s$ corresponding to $\mathbf{X}_s \cap \mathbf{X}'_s$.

# 3. The MO-CBO Problem

In our setting, we assume that the causal relationships encoded in $\mathcal{G}$ are known while the underlying parametrizations, i.e., $\mathbf{F}$ and $P(\mathbf{U})$, can be unknown. This restricted information is denoted as $\langle \mathcal{G}, \mathbf{Y}, \mathbf{X}, \mathbf{C} \rangle$. The assumption is common within the CBO line of work and allows generalization across systems with the same causal structure.

A MO-CBO problem aims to identify intervention set-value pairs $(\mathbf{X}_s, \mathbf{x}_s)$ that offer optimal trade-offs in minimizing (or, maximizing) all target variables in $\mathbf{Y}$. The outcomes of an intervention $\mathrm{do}(\mathbf{X}_s = \mathbf{x}_s)$ are captured as the expected values

$$\mu_i(\mathbf{X}_s, \mathbf{x}_s) := \mathbb{E}_{P(Y_i|\mathrm{do}(\mathbf{X}_s = \mathbf{x}_s))}[Y_i], \qquad (3)$$

where $P(Y_i|\mathrm{do}(\mathbf{X}_s = \mathbf{x}_s))$ denotes the interventional distribution of $Y_i$, for all $i = 1, \ldots, m$. We write $\boldsymbol{\mu}(\mathbf{X}_s, \mathbf{x}_s) = (\mu_1(\mathbf{X}_s, \mathbf{x}_s), \ldots, \mu_m(\mathbf{X}_s, \mathbf{x}_s))^\top$ for the vector notation. Next, we adopt the notion of Pareto optimality to intervention set-value pairs:

**Definition 3.1** (Pareto-optimal intervention set-value pair). Given $\mathcal{S} \subseteq \mathbb{P}(\mathbf{X})$, an intervention set-value pair $(\mathbf{X}_s, \mathbf{x}_s)$ with $\mathbf{X}_s \in \mathcal{S}, \mathbf{x}_s \in \mathcal{D}(\mathbf{X}_s)$ is called *Pareto-optimal for $\mathcal{S}$*, if there is no other intervention set-value pair $(\mathbf{X}'_s, \mathbf{x}'_s)$ with $\mathbf{X}'_s \in \mathcal{S}, \mathbf{x}'_s \in \mathcal{D}(\mathbf{X}'_s)$ such that $\mu_i(\mathbf{X}'_s, \mathbf{x}'_s) \leq \mu_i(\mathbf{X}_s, \mathbf{x}_s)$ for all $1 \leq i \leq m$ and $\mu_i(\mathbf{X}'_s, \mathbf{x}'_s) < \mu_i(\mathbf{X}_s, \mathbf{x}_s)$ for at least one $1 \leq i \leq m$.

**Definition 3.2** (Pareto front for $\mathcal{S}$). The space of all Pareto-optimal intervention set-value pairs for a given $\mathcal{S} \subseteq \mathbb{P}(\mathbf{X})$ is called the *Pareto set for $\mathcal{S}$*, denoted $\mathcal{P}^c_s(\mathcal{S})$. The corresponding *Pareto front for $\mathcal{S}$*, denoted $\mathcal{P}^c_f(\mathcal{S})$, is the $m$-dimensional image of $\mathcal{P}^c_s(\mathcal{S})$ under the objectives $\mu_i, 1 \leq i \leq m$.

We define MO-CBO problems as identifying the Pareto set $\mathcal{P}^c_s(\mathbb{P}(\mathbf{X}))$ which yields the optimal trade-offs among the objectives, represented by the Pareto front $\mathcal{P}^c_f(\mathbb{P}(\mathbf{X}))$.

## 3.1. Decomposition of MO-CBO Problems

We aim to simplify the search space to navigate the discovery of Pareto-optimal intervention set-value pairs.

**Definition 3.3** (Local MO-CBO problem). Let $\mathbf{X}_s \in \mathbb{P}(\mathbf{X})$ be an intervention set. Then, the multi-objective optimization problem defined by the objective functions $\mu_i(\mathbf{X}_s, \cdot) : \mathcal{D}(\mathbf{X}_s) \to \mathbb{R}, \mathbf{x}_s \mapsto \mu_i(\mathbf{X}_s, \mathbf{x}_s), 1 \leq i \leq m$, is called the *local MO-CBO problem w.r.t. $\mathbf{X}_s$*.

The Pareto set of the local MO-CBO problem w.r.t. $\mathbf{X}_s \in \mathbb{P}(\mathbf{X})$ is denoted as $\mathcal{P}^l_s(\mathbf{X}_s)$ and the associated Pareto front as $\mathcal{P}^l_f(\mathbf{X}_s)$. Each local MO-CBO problem corresponds to a standard multi-objective optimization task, solvable with existing methods. The following proposition decomposes MO-CBO problems into such local problems.

**Proposition 3.4.** *Given $\langle \mathcal{G}, \mathbf{Y}, \mathbf{X}, \mathbf{C} \rangle$, let $\mathcal{S} \subseteq \mathbb{P}(\mathbf{X})$ be a non-empty collection of intervention sets. Then, it holds*

$$\mathcal{P}^c_f(\mathcal{S}) \subseteq \bigcup_{s=1}^{|\mathcal{S}|} \mathcal{P}^l_f(\mathbf{X}_s). \qquad (4)$$

*Proof.* See Appendix A. Core idea: We exploit that the space of all intervention set-value pairs is the union of the input spaces of each local problem. $\qquad \square$

Proposition 3.4 allows to match the Pareto-optimal intervention set-value pairs to the Pareto-optimal solutions from the local problems where the intervention set is fixed. Therefore, discovering $\mathcal{P}^c_f(\mathbb{P}(\mathbf{X}))$ requires identifying Pareto-optimal solutions of local MO-CBO problems with respect to all intervention sets $\mathbf{X}_s \in \mathbb{P}(\mathbf{X})$.

# 4. Solving MO-CBO Problems

In this section, we propose our methodology for solving MO-CBO problems, which has been outlined in Figure 2. In summary, we reduce the search space to a subset $\mathcal{S} \subseteq \mathbb{P}(\mathbf{X})$, solve the corresponding local MO-CBO problems w.r.t. each element in $\mathcal{S}$, and extract only Pareto-optimal intervention set-value pairs to construct the Pareto front $\mathcal{P}^c_f(\mathbb{P}(\mathbf{X}))$.

## 4.1. Reducing the Search Space

The complexity of solving the local MO-CBO problems w.r.t. all $\mathbf{X}_s \in \mathbb{P}(\mathbf{X})$ rises exponentially with the number of treatment variables, making this strategy impracticable for most tasks. This section proposes a method to exploit the graph topology to identify a minimal subset $\mathcal{S} \subseteq \mathbb{P}(\mathbf{X})$ with $\mathcal{P}^c_f(\mathbb{P}(\mathbf{X})) = \mathcal{P}^c_f(\mathcal{S})$. Hereby, we generalize the results from Lee & Bareinboim (2018) to the multi-objective setting. All proofs and derivations are given Appendix B. For now, we assume that there are no non-manipulative vari-

ables, i.e., $\mathbf{C} = \varnothing$. At the end of the section, we discuss the general case $\mathbf{C} \neq \varnothing$.

We first reduce the search space by disregarding intervention sets where some variables do not affect the targets:

**Definition 4.1** (Minimal intervention set). A set $\mathbf{X}_s \in \mathbb{P}(\mathbf{X})$ is called a *minimal intervention set* if there exists no subset $\mathbf{X}'_s \subset \mathbf{X}_s$ such that for all $\mathbf{x}_s \in \mathcal{D}(\mathbf{X}_s)$ it holds $\mu_i(\mathbf{X_s}, \mathbf{x}_s) = \mu_i(\mathbf{X}'_s, \mathbf{x}_s[\mathbf{X}'_s]), 1 \leq i \leq m$, for every SCM conforming to $\mathcal{G}$.

We denote the set of minimal intervention sets with $\mathbb{M}_{\mathcal{G}, \mathbf{Y}}$. The following proposition characterizes such sets in a given causal graph $\mathcal{G}$.

**Proposition 4.2.** $\mathbf{X}_s \in \mathbb{P}(\mathbf{X})$ *is a minimal intervention set if and only if it holds* $\mathbf{X}_s \subseteq \mathrm{an}(\mathbf{Y})_{\mathcal{G}_{\overline{\mathbf{X}}_s}}$.

*Proof.* See Appendix B.1. Core idea: The "if" direction shows by contradiction that any non-minimal intervention set cannot consist solely of the ancestors of $\mathbf{Y}$. The "only if" direction is straightforward to prove since variables without an ancestral relationship to $\mathbf{Y}$ are redundant to intervene upon. $\square$

We adapt the notion of possibly-optimal minimal intervention sets (Lee & Bareinboim, 2018) for Pareto-optimality. Intuitively, a minimal intervention set is called possibly Pareto-optimal if it includes a Pareto-optimal intervention set-value pair whose outcome is unattainable with any other intervention set, for at least one SCM conforming to $\mathcal{G}$.

**Definition 4.3** (Possibly Pareto-optimal minimal intervention set). A set $\mathbf{X}_s \in \mathbb{M}_{\mathcal{G}, \mathbf{Y}}$ is called *possibly Pareto-optimal* if, for at least one SCM conforming to $\mathcal{G}$, there exists $\mathbf{x}_s \in \mathcal{D}(\mathbf{X}_s)$ such that $(\mathbf{X}_s, \mathbf{x}_s)$ is Pareto-optimal for $\mathbb{P}(\mathbf{X})$, and for no $\mathbf{X}'_s \in \mathbb{M}_{\mathcal{G}, \mathbf{Y}} \setminus \mathbf{X}_s, \mathbf{x}'_s \in \mathcal{D}(\mathbf{X}'_s)$ it holds $\mu_i(\mathbf{X}'_s, \mathbf{x}'_s) \leq \mu_i(\mathbf{X}_s, \mathbf{x}_s)$, for all $1 \leq i \leq m$.

We denote the set of all possibly Pareto-optimal minimal intervention sets by $\mathbb{O}_{\mathcal{G}, \mathbf{Y}}$. Next, we establish graph-theoretical criteria to identify such sets in a given causal graph. First, the proposition below considers a special case:

**Proposition 4.4.** *If no $Y_i$ is confounded with* $\mathrm{an}(Y_i)_{\mathcal{G}}$ *via unobserved confounders, then* $\mathrm{pa}(\mathbf{Y})_{\mathcal{G}}$ *is the only possibly Pareto-optimal minimal intervention set.*

*Proof.* See Appendix B.2. Core idea: In the absence of unobserved confounding between any $Y_i$ and its ancestors $\mathrm{an}(Y_i)_{\mathcal{G}}$, the average effect of any intervention $\mathrm{do}(\mathbf{X}_s = \mathbf{x}_s)$ can be matched by intervening on $\mathrm{pa}(\mathbf{Y})_{\mathcal{G}}$. $\square$

To characterize possibly Pareto-optimal minimal intervention sets in arbitrary graphs, we extend the following two

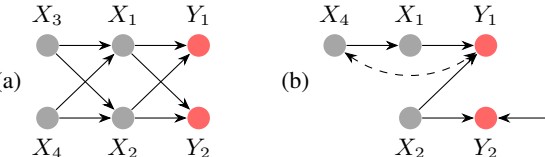

*Figure 3.* Two causal graphs with $\mathbf{X} = \{X_1, X_2, X_3, X_4\}, \mathbf{Y} = \{Y_1, Y_2\}$. (a) No unobserved confounders. (b) The dashed bidirected edge depicts an unobserved confounder between $X_4$ and $Y_1$.

definitions from Lee & Bareinboim (2018) to the multi-objective setting. They aim to identify a region, starting from $\mathbf{Y}$, that is governed by unobserved confounders, along with its outside border that determines the realization of variables within the region.

**Definition 4.5** (Minimal unobserved confounders' territory). Let $\mathcal{H} = \mathcal{G}[\mathrm{An}(\mathbf{Y})_{\mathcal{G}}]$. A set of variables $\mathbf{T}$ in $\mathcal{H}$, with $\mathbf{Y} \subseteq \mathbf{T}$, is called a *UC-territory* for $\mathcal{G}$ w.r.t. $\mathbf{Y}$ if $\mathrm{De}(\mathbf{T})_{\mathcal{H}} = \mathbf{T}$ and $\mathrm{CC}(\mathbf{T})_{\mathcal{H}} = \mathbf{T}$. The UC-territory $\mathbf{T}$ is said to be *minimal*, denoted $\mathbf{T} = \mathrm{MUCT}(\mathcal{G}, \mathbf{Y})$, if no $\mathbf{T}' \subset \mathbf{T}$ is a UC-territory.

**Definition 4.6** (Interventional border). Let us denote $\mathbf{T} = \mathrm{MUCT}(\mathcal{G}, \mathbf{Y})$. Then, $\mathbf{B} = \mathrm{pa}(\mathbf{T})_{\mathcal{G}} \setminus \mathbf{T}$ is called the *interventional border* for $\mathcal{G}$ w.r.t. $\mathbf{Y}$, which we write as $\mathrm{IB}(\mathcal{G}, \mathbf{Y})$.

**Example** We illustrate these two concepts with the causal graphs from Figure 3. In Figure 3 (a), there are no unobserved confounders and thus, it holds $\mathrm{CC}(\mathbf{Y})_{\mathcal{G}} = \mathbf{Y}$ and $\mathrm{De}(\mathbf{Y})_{\mathcal{G}} = \mathbf{Y}$. It follows $\mathrm{MUCT}(\mathcal{G}, \mathbf{Y}) = \{Y_1, Y_2\}$ and $\mathrm{IB}(\mathcal{G}, \mathbf{Y}) = \{X_1, X_2\}$. In Figure 3 (b), we construct the minimal UC-territory, starting from $\mathbf{T} = \mathbf{Y}$, as follows: Since $Y_1$ has an unobserved confounder with $X_4$, we update $\mathbf{T} = \mathrm{CC}(\mathbf{Y})_{\mathcal{G}} = \{Y_1, Y_2, X_4\}$, and thereafter add all the descendants of $X_4$, obtaining $\mathbf{T} = \{Y_1, Y_2, X_4, X_1\}$. Since there are no more unobserved confounders between $\mathbf{T}$ and $\mathrm{An}(\mathbf{Y})_{\mathcal{G}} \setminus \mathbf{T}$, the minimal UC-territory has been found and is given by $\mathrm{MUCT}(\mathcal{G}, \mathbf{Y}) = \{Y_1, Y_2, X_1, X_4\}$ along with $\mathrm{IB}(\mathcal{G}, \mathbf{Y}) = \{X_2, X_3\}$.

Interventional borders can fully determine possibly Pareto-optimal minimal intervention sets, which are described with the following two results.

**Proposition 4.7.** $\mathrm{IB}(\mathcal{G}_{\overline{\mathbf{X}}_s}, \mathbf{Y})$ *is a possibly Pareto-optimal minimal intervention set for any* $\mathbf{X}_s \in \mathbb{P}(\mathbf{X})$.

*Proof.* See Appendix B.2. Core idea: We first prove in Proposition B.2 that $\mathrm{IB}(\mathcal{G}, \mathbf{Y})$ is a possibly Pareto-optimal minimal intervention set by constructing an SCM where $\mathrm{do}(\mathrm{IB}(\mathcal{G}, \mathbf{Y}) = 0)$ is the single best intervention. This construction then easily extends to show that $\mathrm{IB}(\mathcal{G}_{\overline{\mathbf{X}}_s}, \mathbf{Y})$ can also represent the single optimal intervention. $\square$

Next, we can finally characterize possibly Pareto-optimal minimal intervention sets:

**Theorem 4.8.** *A set $\mathbf{X}_s \in \mathbb{P}(\mathbf{X})$ is a possibly Pareto-optimal minimal intervention set if and only if it holds $\mathrm{IB}(\mathcal{G}_{\overline{\mathbf{X}}_s}, \mathbf{Y}) = \mathbf{X}_s$.*

*Proof.* See Appendix B.2. Core idea: The "if" statement is a special case of Proposition 4.7. We prove the "only if" direction by showing that intervening on $\mathrm{IB}(\mathcal{G}_{\overline{\mathbf{X}}_s}, \mathbf{Y})$ is at least as optimal as intervening on $\mathbf{X}_s$, $\qquad\square$

**Corollary 4.9.** *Let $\mathbf{X}_s \in \mathbb{P}(\mathbf{X})$ and $\mathbf{X}'_s = \mathrm{IB}(\mathcal{G}_{\overline{\mathbf{X}}_s}, \mathbf{Y})$. For any $\mathbf{x}_s \in \mathcal{D}(\mathbf{X}_s)$ there exist $\mathbf{x}'_s \in \mathcal{D}(\mathbf{X}'_s)$ such that it holds $\mu(\mathbf{X}'_s, \mathbf{x}'_s) \leq \mu(\mathbf{X}_s, \mathbf{x}_s)$, for all $1 \leq i \leq m$.*

Corollary 4.9 is a direct result from the proof of Theorem 4.8. In this setting, it is easy to construct an SCM, conforming to $\mathcal{G}$, for which strict inequality holds in at least one component. Finally, we show that it suffices to only consider possibly Pareto-optimal minimal intervention sets to solve MO-CBO problems.

**Theorem 4.10.** *It holds $\mathcal{P}^{\mathsf{c}}_f(\mathbb{P}(\mathbf{X})) = \mathcal{P}^{\mathsf{c}}_f(\mathbb{O}_{\mathcal{G},\mathbf{Y}})$.*

*Proof.* $\subseteq$: Assume $\mathcal{P}^{\mathsf{c}}_f(\mathbb{P}(\mathbf{X})) \not\subseteq \mathcal{P}^{\mathsf{c}}_f(\mathbb{O}_{\mathcal{G},\mathbf{Y}})$. Then, there exists $\mathbf{z} \in \mathbb{R}^m$, with $\mathbf{z} = \boldsymbol{\mu}(\mathbf{X}_s, \mathbf{x}_s)$ for some intervention set-value pair $(\mathbf{X}_s, \mathbf{x}_s)$, such that $\mathbf{z} \in \mathcal{P}^{\mathsf{c}}_f(\mathbb{P}(\mathbf{X}))$ and $\mathbf{z} \notin \mathcal{P}^{\mathsf{c}}_f(\mathbb{O}_{\mathcal{G},\mathbf{Y}})$. If $\mathbf{X}_s \in \mathbb{O}_{\mathcal{G},\mathbf{Y}}$, it follows that $(\mathbf{X}_s, \mathbf{x}_s)$ is not Pareto-optimal for $\mathbb{O}_{\mathcal{G},\mathbf{Y}}$, which is a contradiction since it is Pareto-optimal for $\mathbb{P}(\mathbf{X})$. Conversely, if $\mathbf{X}_s \in \mathbb{P}(\mathbf{X}) \backslash \mathbb{O}_{\mathcal{G},\mathbf{Y}}$, we set $\mathbf{X}'_s = \mathrm{IB}(\mathcal{G}_{\overline{\mathbf{X}}_s}, \mathbf{Y})$ and from Corollary 4.9, we infer that, for some SCM conforming to $\mathcal{G}$, there exists $\mathbf{x}'_s \in \mathcal{D}(\mathbf{X}'_s)$ with $\mu(\mathbf{X}'_s, \mathbf{x}'_s) \leq \mu(\mathbf{X}_s, \mathbf{x}_s)$, for all $1 \leq i \leq m$, and $\mu(\mathbf{X}'_s, \mathbf{x}'_s) < \mu(\mathbf{X}_s, \mathbf{x}_s)$, for at least one $1 \leq i \leq m$. This results in $\mathbf{z} \notin \mathcal{P}^{\mathsf{c}}_f(\mathbb{P}(\mathbf{X}))$, which is a contradiction.

$\supseteq$: Assume $\mathcal{P}^{\mathsf{c}}_f(\mathbb{O}_{\mathcal{G},\mathbf{Y}}) \not\subseteq \mathcal{P}^{\mathsf{c}}_f(\mathbb{P}(\mathbf{X}))$. Then, there exists $\mathbf{z} \in \mathbb{R}^m$, with $\mathbf{z} = \boldsymbol{\mu}(\mathbf{X}_s, \mathbf{x}_s)$, such that $\mathbf{z} \in \mathcal{P}^{\mathsf{c}}_f(\mathbb{O}_{\mathcal{G},\mathbf{Y}})$ and $\mathbf{z} \notin \mathcal{P}^{\mathsf{c}}_f(\mathbb{P}(\mathbf{X}))$. There exists some $\mathbf{X}'_s \in \mathbb{P}(\mathbf{X}) \backslash \mathbb{O}_{\mathcal{G},\mathbf{Y}}$, $\mathbf{x}_s \in \mathcal{D}(\mathbf{X}'_s)$ such that $(\mathbf{X}'_s, \mathbf{x}'_s)$ is Pareto optimal and for which it holds $\mu(\mathbf{X}'_s, \mathbf{x}'_s) \leq \mu(\mathbf{X}_s, \mathbf{x}_s)$, for all $1 \leq i \leq m$, and $\mu(\mathbf{X}'_s, \mathbf{x}'_s) < \mu(\mathbf{X}_s, \mathbf{x}_s)$, for at least one $1 \leq i \leq m$. Since $\mathbf{X}'_s$ is not possibly Pareto-optimal, we infer from Corollary 4.9 that for $\mathbf{X}''_s = \mathrm{IB}(\mathcal{G}, \mathbf{Y})$ there exists $\mathbf{x}''_s \in \mathcal{D}(\mathbf{X}''_s)$ such that $\mu(\mathbf{X}''_s, \mathbf{x}''_s) \leq \mu(\mathbf{X}'_s, \mathbf{x}'_s)$, for all $1 \leq i \leq m$. Hence, it holds $\boldsymbol{\mu}(\mathbf{X}''_s, \mathbf{x}''_s) \in \mathcal{P}^{\mathsf{c}}_f(\mathbb{P}(\mathbf{X}))$, which is a contradiction to $\mathbf{z} \in \mathcal{P}^{\mathsf{c}}_f(\mathbb{O}_{\mathcal{G},\mathbf{Y}})$ since $\mathbf{X}''_s \in \mathbb{O}_{\mathcal{G},\mathbf{Y}}$. $\qquad\square$

Using Theorem 4.10, we reduce the search space of MO-CBO problems to $\mathcal{S} = \mathbb{O}_{\mathcal{G},\mathbf{Y}}$.

**Example** We illustrate the search space reduction with the causal graphs from Figure 3. Note that in both cases it holds $\mathbb{P}(\mathbf{X}) = 2^{|\mathbf{X}|} = 16$. In Figure 3 (a), there are no unobserved confounders, and it follows $\mathbb{O}_{\mathcal{G},\mathbf{Y}} = \mathrm{pa}(\mathbf{Y})_{\mathcal{G}} = \{X_1, X_2\}$.

---

**Algorithm 1** Our MO-CBO algorithm

**Input:** $\langle \mathcal{G}, \mathbf{Y}, \mathbf{X}, \mathbf{C} \rangle$, $\mathcal{S} \in \{\mathbb{O}_{\mathcal{G},\mathbf{Y}}, \mathbb{M}_{\mathcal{G},\mathbf{Y}}, \mathbb{P}(\mathbf{X})\}$, data $\mathcal{D}$, batch size $B$, number of iterations $N$
**Output:** $\mathcal{P}^{\mathsf{c}}_s(\mathcal{S})$, $\mathcal{P}^{\mathsf{c}}_f(\mathcal{S})$
Initialize the dataset $\mathcal{D}_0 = \mathcal{D}$
**for** $s = 1$ **to** $|\mathcal{S}|$ **do**
    Fit surrogates $\tilde{\mu}_i(\mathbf{X}_\mathbf{s}, \cdot)$ with $\mathcal{D}_0$, $i = 1, \dots, m$
    Approximate $\mathcal{P}^{\mathsf{l}}_s(\mathbf{X}_s)$ and $\mathcal{P}^{\mathsf{l}}_f(\mathbf{X}_s)$ using $\tilde{\mu}_1, \dots, \tilde{\mu}_m$
**end for**
**for** $n = 1$ **to** $N$ **do**
    **for** $s = 1$ **to** $|\mathcal{S}|$ **do**
        Select batch $\mathbf{B}_s = \{\mathbf{x}^b_s\}^B_{b=1}$ via Equation (2)
    **end for**
    Select batch $\mathbf{B}_{\hat{s}}$ from $\{\mathbf{B}_1, \dots, \mathbf{B}_{|\mathcal{S}|}\}$ via Equation (6)
    Intervene on $\mathbf{X}_{\hat{s}}$ with $\mathbf{B}_{\hat{s}}$
    Augment $\mathcal{D}_n = \mathcal{D}_{n-1} \cup \{(\mathbf{X}_{\hat{s}}, \mathbf{x}^b_{\hat{s}}), \boldsymbol{\mu}(\mathbf{X}_{\hat{s}}, \mathbf{x}^b_{\hat{s}})\}^B_{b=1}$
    Update surrogates $\tilde{\mu}_i(\mathbf{X}_{\hat{s}}, \cdot)$ with $\mathcal{D}_n$, $i = 1, \dots, m$
    Approximate $\mathcal{P}^{\mathsf{l}}_s(\mathbf{X}_{\hat{s}})$ and $\mathcal{P}^{\mathsf{l}}_f(\mathbf{X}_{\hat{s}})$ using $\tilde{\mu}_1, \dots, \tilde{\mu}_m$
**end for**
Compute $\mathcal{P}^{\mathsf{l}}_s(\mathbf{X}_s)$, $\mathcal{P}^{\mathsf{l}}_f(\mathbf{X}_s)$ from $\mathcal{D}_N$, $s = 1, \dots, |\mathcal{S}|$
Compute $\mathcal{P}^{\mathsf{c}}_s(\mathcal{S})$ and $\mathcal{P}^{\mathsf{c}}_f(\mathcal{S})$

---

In Figure 3 (b), the intervention sets $\{X_1, X_2, X_3\}$ and $\{X_2, X_3\}$ satisfy the condition from Theorem 4.8, and thus, $\mathbb{O}_{\mathcal{G},\mathbf{Y}} = \{\{X_2, X_3\}, \{X_1, X_2, X_3\}\}$.

We now consider the more general case with $\mathbf{C} \neq \varnothing$, where non-manipulative variables can be present. The definitions for the minimal intervention set and the possibly Pareto-optimal minimal intervention set are a straight-forward extension. Lee & Bareinboim (2019a) propose a projection $\mathcal{G} \rightarrow \mathcal{G}[\mathbf{V} \backslash \mathbf{C}]$ which preserves the distribution of the underlying SCM. Given such a projection, we can identify the possibly Pareto-optimal minimal intervention sets in $\langle \mathcal{G}, \mathbf{Y}, \mathbf{X}, \mathbf{C} \rangle$ by applying Theorem 4.8 to $\langle \mathcal{G}[\mathbf{V} \backslash \mathbf{C}], \mathbf{Y}, \mathbf{X} \rangle$.

### 4.2. Solving the Local Problems

We propose our algorithm to solve MO-CBO problems[1], for which the procedure is summarized in Algorithm 1. It assumes a known causal graph $\langle \mathcal{G}, \mathbf{Y}, \mathbf{X}, \mathbf{C} \rangle$, prior data $\mathcal{D}$, and a set $\mathcal{S} \in \{\mathbb{O}_{\mathcal{G},\mathbf{Y}}, \mathbb{M}_{\mathcal{G},\mathbf{Y}}, \mathbb{P}(\mathbf{X})\}$ that specifies which local problems to consider. The idea is to alternately solve the local MO-CBO problems using the MOBO algorithm DGEMO.

More specifically, the algorithm operates as follows: For each local MO-CBO problem w.r.t. $\mathbf{X}_s \in \mathcal{S}$, it first fits the surrogate model to the objectives $\mu_i(\mathbf{X}_s, \cdot)$, $1 \leq i \leq m$, via independent Gaussian processes. Based on the means of

---

[1]The full implementation of our algorithm is available at https://github.com/ShriyaBhatija/MO-CBO

*Table 1.* The generational distances, averaged across 10 random seeds. Lower values indicate closer approximation to $\mathcal{P}_f^{\mathsf{c}}(\mathbb{P}(\mathbf{X}))$.

| Problem | ParEGO | MOEA/D-EGO | TSEMO | $q$NEHVI | DGEMO | MO-CBO (ours) |
|---|---|---|---|---|---|---|
| SYNTHETIC-1 | 0.30 | 0.38 | 0.16 | 0.27 | 0.14 | **0.14** |
| SYNTHETIC-2 | 12.43 | 11.45 | 8.65 | 7.98 | 6.79 | **2.80** |
| HEALTH | 0.09 | 0.20 | 0.08 | 0.12 | 0.07 | **0.06** |
| CREDIT APPROVAL | 0.14 | 0.12 | 0.06 | 0.08 | 0.06 | **0.05** |

*Table 2.* The inverted generational distances, averaged across 10 random seeds. Lower values indicate a more diverse coverage of solutions approximate to $\mathcal{P}_f^{\mathsf{c}}(\mathbb{P}(\mathbf{X}))$.

| Problem | ParEGO | MOEA/D-EGO | TSEMO | $q$NEHVI | DGEMO | MO-CBO (ours) |
|---|---|---|---|---|---|---|
| SYNTHETIC-1 | 3.63 | 3.24 | 2.82 | 2.45 | 2.57 | **1.40** |
| SYNTHETIC-2 | 7.43 | 7.70 | 4.78 | 5.49 | 4.50 | **0.87** |
| HEALTH | 0.15 | 0.16 | 0.05 | 0.10 | 0.05 | **0.02** |
| CREDIT APPROVAL | 0.24 | 0.28 | **0.08** | 0.21 | 0.09 | **0.08** |

the Gaussian process posteriors, approximations of $\mathcal{P}_s^{\mathsf{l}}(\mathbf{X}_s)$ and $\mathcal{P}_f^{\mathsf{l}}(\mathbf{X}_s)$ are computed utilizing the Pareto discovery approach from DGEMO. After this initial step, the most promising intervention set is selected for batch evaluation at each iteration. The dataset is then augmented with the newly evaluated batch of samples. For the corresponding local problem, we again update the surrogate model and the Pareto set and front approximations. After completing all iterations, the algorithm identifies the final Pareto sets and fronts for each local MO-CBO problem using the collected objective function evaluations $\mathcal{D}_N$. Thereafter, it is easy to construct $\mathcal{P}_s^{\mathsf{c}}(\mathcal{S})$ and $\mathcal{P}_f^{\mathsf{c}}(\mathcal{S})$, see Section 4.3.

**Batch Selection** For the local MO-CBO problem w.r.t. $\mathbf{X}_s \in \mathcal{S}$, let $\mathcal{R}_1(\mathbf{X}_s), \ldots, \mathcal{R}_K(\mathbf{X}_s) \subseteq \mathcal{D}(\mathbf{X}_s)$ denote the identified diversity regions from DGEMO, discussed in Section 2.1. Our algorithm seeks to balance the exploration of Pareto fronts from multiple local MO-CBO problems, but evaluating all $\mathbf{B}_s$, $s = 1, \ldots, |\mathcal{S}|$, during a single iteration, is an inefficient strategy. Instead, we select the batch with the most promising hypervolume improvement at each iteration. To this end, we introduce the term *relative hypervolume improvement*, defined as

$$\mathrm{RHVI}(\boldsymbol{\mu}(\mathbf{X}_s, \mathbf{B}_s), \mathcal{P}_f^{\mathsf{l}}(\mathbf{X_s})) = \frac{\mathrm{HVI}(\boldsymbol{\mu}(\mathbf{X}_s, \mathbf{B}_s), \mathcal{P}_f^{\mathsf{l}}(\mathbf{X_s}))}{\mathcal{H}(\mathcal{P}_f^{\mathsf{l}}(\mathbf{X}_s))}. \quad (5)$$

As the name suggests, relative hypervolume improvement is a normalized measure of improvement, enabling the assessment of batch evaluations across different intervention sets. Given $\mathbf{B}_1, \ldots, \mathbf{B}_{|\mathcal{S}|}$, we propose the following batch selection strategy for our MO-CBO algorithm:

$$\mathbf{B}_{\hat{s}} = \underset{\mathbf{B}_s \in \{\mathbf{B}_1, \ldots, \mathbf{B}_{|\mathcal{S}|}\}}{\arg\max} \mathrm{RHVI}(\boldsymbol{\mu}(\mathbf{X}_s, \mathbf{B}_s), \mathcal{P}_f^{\mathsf{l}}(\mathbf{X_s})). \quad (6)$$

Overall, the proposed batch selection is designed to alternately advance the Pareto fronts $\mathcal{P}_f^{\mathsf{l}}(\mathbf{X}_1), \ldots, \mathcal{P}_f^{\mathsf{l}}(\mathbf{X}_{|\mathcal{S}|})$.

### 4.3. Building the Pareto Front

After Algorithm 1 has computed the Pareto sets and Pareto fronts of the local problems, its final step is to simply extract Pareto-optimal points from $\bigcup_{s=1}^{|\mathcal{S}|} \mathcal{P}_s^{\mathsf{l}}(\mathbf{X}_s)$, as justified by Proposition 3.4. This yields Pareto-optimal intervention set-value pairs $\mathcal{P}_s^{\mathsf{c}}(\mathbb{P}(\mathbf{X}))$ and their corresponding Pareto front $\mathcal{P}_f^{\mathsf{c}}(\mathbb{P}(\mathbf{X}))$.

## 5. Experiments

We evaluate our MO-CBO algorithm with $\mathcal{S} = \mathbb{O}_{\mathcal{G}, \mathbf{Y}}$ on the causal graphs shown in Figure 3 (a) (SYNTHETIC-1), Figure 3 (b) (SYNTHETIC-2), Figure 1 (HEALTH), and an additional CREDIT APPROVAL example. We cover both synthetic and real-world scenarios. The full description of the underlying SCMs is given in Appendix C. We assume to have an initial dataset $\mathcal{D} = \{((\mathbf{X}_s, \mathbf{x}_s^k), \boldsymbol{\mu}(\mathbf{X}_s, \mathbf{x}_s^k))\}_{k=1, s=1}^{K, |\mathcal{S}|}$ with $K = 5$ samples per intervention set. The batch size is set to 5. For reproducibility, all experiments are run across 10 random seeds, resulting in varying initializations of $\mathcal{D}$.

**Baselines** To the best of our knowledge, there exists no other multi-objective optimization method in the literature that can leverage the causal structure. As baselines, we therefore apply some of the most prominent MOBO algorithms such as ParEGO, MOEA/D-EGO, TSEMO, $q$NEHVI, and DGEMO (see the literature review). They intervene on all treatment variables simultaneously and thus the objective functions are $\mu_i(\mathbf{X}, \cdot) : \mathcal{D}(\mathbf{X}) \to \mathbb{R}$, $\mathbf{x} \mapsto \mu(\mathbf{X}, \mathbf{x})$, $i = 1, \ldots, m$. Notably, $\mathcal{P}_f^{\mathsf{c}}(\mathbb{P}(\mathbf{X}))$ contains at least as optimal outcomes as $\mathcal{P}_f^{\mathsf{l}}(\mathbf{X})$.

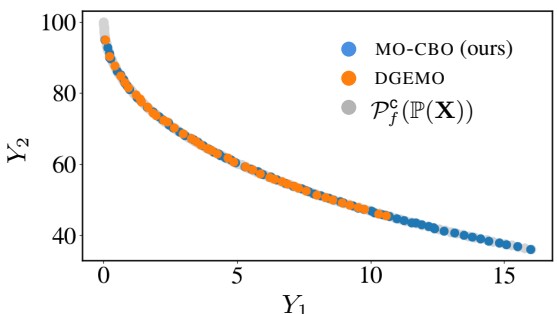

*Figure 4.* SYNTHETIC-1. Pareto front approximations from MO-CBO (ours) and DGEMO. Our method offers a higher coverage of $\mathcal{P}_f^{\mathsf{c}}(\mathbb{P}(\mathbf{X}))$.

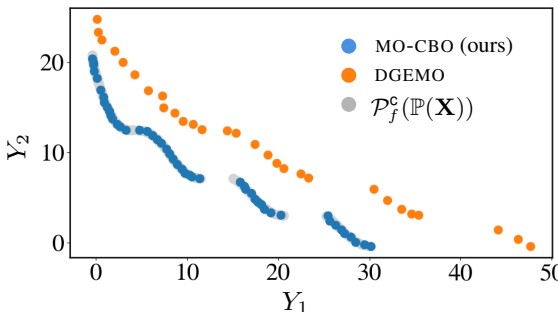

*Figure 5.* SYNTHETIC-2. Pareto front approximations from MO-CBO (ours) and DGEMO. Our approach tightly fits $\mathcal{P}_f^{\mathsf{c}}(\mathbb{P}(\mathbf{X}))$.

**Evaluation** We assess the quality of the resulting Pareto fronts by measuring their proximity to $\mathcal{P}_f^{\mathsf{c}}(\mathbb{P}(\mathbf{X}))$ using the generational distance (GD) (Schutze et al., 2012). The GD is defined as the average distance from any point in the approximated front to its closest point on the ground-truth front. Moreover, we calculate the diversity of the identified solutions using the inverted generational distance (IGD) (Schutze et al., 2012). The IGD represents the average distance from any point in the ground-truth front to its closest point on the approximated front. The mathematical definitions are given in Appendix D. We present the performance metrics from our experiments in Table 1 and Table 2.

We run each experiment until a predefined cost budget is exhausted, assuming each intervention comes at a certain cost. The cost structure is detailed in Appendix D. In this section, we will only present the visual results from the experiments with DGEMO, with corresponding plots for the other baselines provided in Appendix D.

### 5.1. Synthetic Problems

**SYNTHETIC-1** As previously discussed in Section 4.1, it holds $\mathbb{O}_{\mathcal{G},\mathbf{Y}} = \{\{X_1, X_2\}\}$. We observe that the generational distances in Table 1 are negligible for MO-CBO and all baseline algorithms. This is also supported by the Pareto front approximations shown in Figure 4 where solutions found by both MO-CBO and DGEMO closely match $\mathcal{P}_f^{\mathsf{c}}(\mathbb{P}(\mathbf{X}))$. Similar results are observed for the other baselines and are presented in Appendix D. Theoretically, these findings are expected, as $\mu(\mathbf{X}, \mathbf{x}) = \mu(\mathbb{O}_{\mathcal{G},\mathbf{Y}}, \mathbf{x}[\mathbb{O}_{\mathcal{G},\mathbf{Y}}])$ guarantees that the baselines can reach the same solutions as MO-CBO. Furthermore, we observe that our method offers a better coverage of $\mathcal{P}_f^{\mathsf{c}}(\mathbb{P}(\mathbf{X}))$, a result confirmed by its lower inverted generational distance in Table 2. The improvement likely stems from avoiding unnecessary interventions on $X_3$ and $X_4$, allowing for more exploratory interventions on $\mathbb{O}_{\mathcal{G},\mathbf{Y}}$ within the same budget.

**SYNTHETIC-2** In this setting, an unobserved confounder exists between $Y_1$ and $X_4$, placing SYNTHETIC-2 in the general case where hidden confounders may influence target variables through their ancestors. Consequently, it holds $\mathbb{O}_{\mathcal{G},\mathbf{Y}} = \{\{X_2, X_3\}, \{X_1, X_2, X_3\}\}$. The Pareto front approximations are illustrated in Figure 5, demonstrating that while the baseline method DGEMO fails to identify solutions on $\mathcal{P}_f^{\mathsf{c}}(\mathbb{P}(\mathbf{X}))$, MO-CBO does indeed discover them. Similar observations are seen for the other baselines, see Appendix D. Further experiments reveal that only interventions on $\{X_2, X_3\}$ can yield Pareto-optimal solutions in $\mathcal{P}_f^{\mathsf{c}}(\mathbb{P}(\mathbf{X}))$. This can be explained as follows: The baseline strategy disrupts the causal path $X_4 \rightarrow X_1 \rightarrow Y_1$, letting the unobserved confounder influence $Y_1$ without propagating through the aforementioned path. In contrast, our approach allows interventions on $\{X_2, X_3\}$, preserving this causal structure. This distinction is crucial as the structural assignment of $Y_1$ includes the term $-X_1 \cdot X_2 \cdot U/2$, with $U$ denoting the unobserved confounder (all structural equations are specified in Appendix C). Not intervening on $X_1$, causes this term to always be negative, yielding lower function values for $Y_1$. However, if we do intervene on $X_1$, it is positive with probability $0.5$, causing higher values for $Y_1$ in the averaged outcomes.

### 5.2. Real-World Problems

**HEALTH** We revisit the causal graph from Figure 1, which is based on real-world causal relationships in the healthcare setting (Ferro et al., 2015). For patients sensitive to Statin medication, one might aim to minimize both Statin usage and PSA levels simultaneously. There is only one possibly Pareto-optimal minimal intervention set, $\mathbb{O}_{\mathcal{G},\mathbf{Y}} = \{\{\text{BMI}, \text{Aspirin}\}\}$. Similarly to SYNTHETIC-1, both our MO-CBO algorithm and the MOBO baselines identify Pareto-optimal solutions, while the baselines tend to produce sparser approximations of $\mathcal{P}_f^{\mathsf{c}}(\mathbb{P}(\mathbf{X}))$. See Figure 6 for the results obtained using DGEMO.

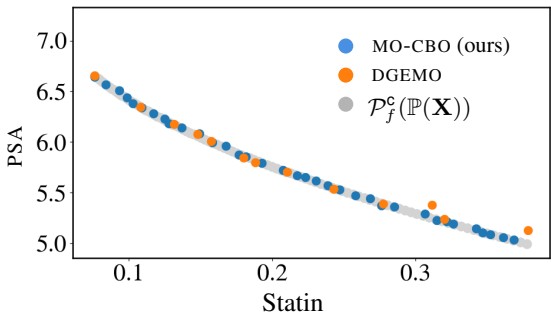

*Figure 6.* HEALTH. Pareto front approximations from MO-CBO (ours) and DGEMO. Our approach yields a better coverage of $\mathcal{P}_f^{\mathtt{c}}(\mathbb{P}(\mathbf{X}))$.

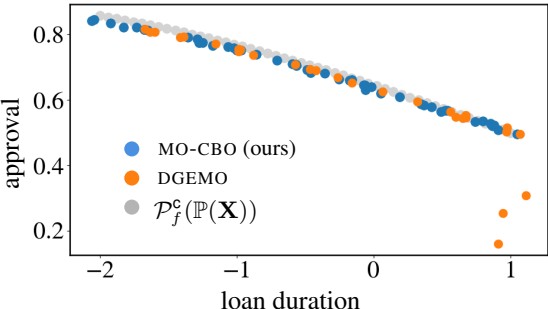

*Figure 7.* CREDIT APPROVAL. Pareto front approximations from MO-CBO (ours) and DGEMO. Our approach yields a better coverage of $\mathcal{P}_f^{\mathtt{c}}(\mathbb{P}(\mathbf{X}))$.

**CREDIT APPROVAL** We consider a causal system which models the credit approval probability as a function of demographic and financial variables, see Appendix D for the SCM specifications. The model is inspired by the German Credit UCI dataset (Murphy, 1994), with causal dependencies adapted from Karimi et al. (2020). Our objective is to maximize the probability of credit approval and the received loan duration (measured as a deviation from the mean). There are no unobserved confounders, resulting in $\mathbb{O}_{\mathcal{G}, \mathbf{Y}} = \{\{\text{loan amount}, \text{income}, \text{savings}\}\}$. Similarly to before, we observe that our MO-CBO algorithm can yield a more dense representation of $\mathcal{P}_f^{\mathtt{c}}(\mathbb{P}(\mathbf{X}))$ compared to some baselines.

## 6. Conclusion

This paper introduces MO-CBO as a new problem class for optimizing multiple target variables within a known causal graph. We show that any MO-CBO problem can be decomposed into local problems, and propose theoretical analyses to identify a minimal collection of such local problems guaranteed to contain all Pareto-optimal solutions. Finally, we present our MO-CBO algorithm that explores this reduced

search space to identify such solutions. Notably, we have observed that traditional multi-objective optimization is simply misspecified for causal systems and can therefore yield suboptimal outcomes. More specifically, our experimental results reveal two distinct scenarios: In the absence of unobserved confounders between the targets and their ancestors, both our MO-CBO algorithm and the MOBO baselines recover optimal solutions, albeit our method can offer a higher solution diversity. In the contrasting scenario, the MOBO baselines can lead to suboptimal solutions, while our method remains effective. These observations align with our theoretical findings.

The search space reduction in Section 4.1 requires prior causal knowledge, which is a notable limitation of our approach. In some domains, such knowledge is accessible through experimental studies (Blomqvist et al., 2020), and when unavailable, could potentially be inferred using methods from causal discovery (Zanga et al., 2022). Moreover, in our current implementation, the surrogate model assumes independent outcomes, overlooking shared endogenous confounders. Future work could enhance sample efficiency by integrating multi-task Gaussian processes to capture shared information across treatment variables. Other directions for future research include the adaptation of existing CBO variants to the multi-objective setting.

## Acknowledgements

This research was supported by the Bavarian State Ministry of Education, Science and the Arts, the Engineering and Physical Sciences Research Council [EP/S023917/1], and the Alan Turing Institute's studentship scheme.

## Impact Statement

This paper presents work whose goal is to advance the field of Machine Learning. There are many potential societal consequences of our work, none which we feel must be specifically highlighted here. We further emphasize that the health-related causal graph is a simplified, illustrative example with no ethical concerns or implications for clinical practice.

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

## A. Decomposition of MO-CBO Problems

Recall the definition of the local MO-CBO problems.

**Definition 3.3** (Local MO-CBO problem). Let $\mathbf{X}_s \in \mathbb{P}(\mathbf{X})$ be an intervention set. Then, the multi-objective optimisation problem defined by the objective functions $\mu_i(\mathbf{X}_s, \cdot) : \mathcal{D}(\mathbf{X}_s) \to \mathbb{R}$, $\mathbf{x}_s \mapsto \mu(\mathbf{X}_s, \mathbf{x}_s)$, $1 \leq i \leq m$, is called *local* MO-CBO *problem w.r.t.* $\mathbf{X}_s$.

For the local MO-CBO problem w.r.t. $\mathbf{X}_s \in \mathbb{P}(\mathbf{X})$, we denote its Pareto set as $\mathcal{P}_s^{\mathsf{l}}(\mathbf{X}_s)$ and the associated Pareto front as $\mathcal{P}_f^{\mathsf{l}}(\mathbf{X}_s)$.

**Proposition 3.4.** *Given* $\langle \mathcal{G}, \mathbf{Y}, \mathbf{X}, \mathbf{C} \rangle$*, let* $\mathcal{S} \subseteq \mathbb{P}(\mathbf{X})$ *be a non-empty collection of intervention sets. Then, it holds*

$$\mathcal{P}_f^{\mathsf{c}}(\mathcal{S}) \subseteq \bigcup_{s=1}^{|\mathcal{S}|} \mathcal{P}_f^{\mathsf{l}}(\mathbf{X}_s). \tag{7}$$

*Proof.* Assume for contradiction that $\mathcal{P}_f^{\mathsf{c}}(\mathcal{S}) \not\subseteq \bigcup_{s=1}^{|\mathcal{S}|} \mathcal{P}_f^{\mathsf{l}}(\mathbf{X}_s)$. Then, there exists some $\mathbf{z} \in \mathbb{R}^m$ such that $\mathbf{z} \in \mathcal{P}_f^{\mathsf{c}}(\mathcal{S})$ and $\mathbf{z} \notin \mathcal{P}_f^{\mathsf{l}}(\mathbf{X}_s)$ for all $s = 1, \ldots, |\mathcal{S}|$. For some intervention set $\mathbf{X}_s' \in \mathcal{S}$ and intervention value $\mathbf{x}_s' \in \mathcal{D}(\mathbf{X}_s')$, it holds $\mathbf{z} = (\mu_1(\mathbf{X}_s', \mathbf{x}_s'), \ldots, \mu_m(\mathbf{X}_s', \mathbf{x}_s'))$. Let $\mathcal{P}_s^{\mathsf{l}}(\mathbf{X}_1), \ldots, \mathcal{P}_s^{\mathsf{l}}(\mathbf{X}_{|\mathcal{S}|})$ be the Pareto sets of the associated local MO-CBO problems w.r.t. $\mathbf{X}_1, \ldots, \mathbf{X}_{|\mathcal{S}|}$, respectively. Since $\mathbf{z} \notin \mathcal{P}_f^{\mathsf{l}}(\mathbf{X}_s')$, it follows $\mathbf{x}_s' \notin \mathcal{P}_s^{\mathsf{l}}(\mathbf{X}_s')$, i.e. $\mathbf{x}_s'$ is not Pareto-optimal in the local MO-CBO problem w.r.t. $\mathbf{X}_s'$. Thus, there exists another intervention value $\mathbf{x}_s'' \in \mathcal{D}(\mathbf{X}_s')$ such that $\mu_i(\mathbf{X}_s', \mathbf{x}_s') \geq \mu_i(\mathbf{X}_s', \mathbf{x}_s'')$ for all $i$ and $\mu_i(\mathbf{X}_s', \mathbf{x}_s') > \mu_i(\mathbf{X}_s', \mathbf{x}_s'')$ for at least one $1 \leq i \leq m$. In other words, the intervention set-value pair $(\mathbf{X}_s', \mathbf{x}_s')$ is not Pareto-optimal for $\mathcal{S}$ since it is dominated by $(\mathbf{X}_s', \mathbf{x}_s'')$. Therefore, $\mathbf{z} \notin \mathcal{P}_f^{\mathsf{c}}(\mathcal{S})$ which is a contradiction. $\square$

## B. Reducing the Search Space

Lee & Bareinboim (2018) leverage the graph topology of an SCM to identify intervention sets that are redundant to consider in any optimisation scheme. Their formalism exploits the rules of do-calculus to identify invariances and partial-orders among intervention sets, in order to obtain those sets that could potentially yield optimal outcomes for a given graph. To take advantage of their ideas for this paper, the relevant concepts and their theoretical properties must be extended to accommodate multi-target settings, which will be the focus of this section.

Let $\langle \mathbf{V}, \mathbf{U}, \mathbf{F}, P(\mathbf{U}) \rangle$ denote an SCM and $\mathcal{G}$ its associated acyclic graph that encodes the underlying causal mechanisms. Recall that we assume $\mathbf{C} = \varnothing$, i.e., there are no non-manipulative variables. In this section, we require the notation $\mathbb{E}_{P(\mathbf{W}|\mathrm{do}(\mathbf{X}_s = \mathbf{x}_s))}[\mathbf{W}] := \mathbb{E}[\mathbf{W}|\mathrm{do}(\mathbf{X}_s = \mathbf{x}_s)]$ for sets $\mathbf{X}_s \subseteq \mathbf{X}$, $\mathbf{W} \subseteq \mathbf{V}$.

### B.1. Equivalence of Intervention Sets

As a first step, we establish invariances within $\mathbb{P}(\mathbf{X})$ in regards to the effects of intervention sets on the target variables. Recall the following definition from the main part of the paper.

**Definition 4.1** (Minimal intervention set). A set $\mathbf{X}_s \in \mathbb{P}(\mathbf{X})$ is called a *minimal intervention set* if, for every SCM conforming to $\mathcal{G}$, there exists no subset $\mathbf{X}_s' \subset \mathbf{X}_s$ such that for all $\mathbf{x}_s \in \mathcal{D}(\mathbf{X}_s)$ it holds $\mu(\mathbf{X_s}, \mathbf{x}_s) = \mu(\mathbf{X}_s', \mathbf{x}_s[\mathbf{X}_s'])$, for all $1 \leq i \leq m$.

We denote the set of minimal intervention sets with $\mathbb{M}_{\mathcal{G}, \mathbf{Y}}$. In other words, no subset of a minimal intervention set can achieve the same expected outcome on $\mathbf{Y}$. Intervention sets, that are not *minimal* in the sense of Definition 4.1, are redundant to consider in any optimization task.

**Proposition 4.2.** $\mathbf{X}_s \in \mathbb{P}(\mathbf{X})$ *is a minimal intervention set if and only if it holds* $\mathbf{X}_s \subseteq \mathrm{an}(\mathbf{Y})_{\mathcal{G}_{\overline{\mathbf{X}}_s}}$.

*Proof.* (If) Let $\mathbf{x}_s \in \mathcal{D}(\mathbf{X}_s)$ be any intervention value. Assume that there is a subset $\mathbf{X}_s' \subset \mathbf{X}_s$ such that for all SCMs conforming to $\mathcal{G}$ it holds $\mathbb{E}[Y_i|\mathrm{do}(\mathbf{X}_s = \mathbf{x}_s)] = \mathbb{E}[Y_i|\mathrm{do}(\mathbf{X}_s' = \mathbf{x}_s[\mathbf{X}_s'])]$ for all $1 \leq i \leq m$. For the sake of contradiction, assume $\mathbf{X}_s \subseteq \mathrm{an}(\mathbf{Y})_{\mathcal{G}_{\overline{\mathbf{X}}_s}}$. Consider an SCM with real-valued variables where each $V \in \mathbf{V}$ is associated with its own binary exogenous variable $U_V$ with $P(U_V = 1) = 0.5$. Let the function of an endogenous variable be the sum of values of its parents. Then, there exists a directed path from $\mathbf{X}_s \backslash \mathbf{X}_s'$ to some $Y_i$ without passing $\mathbf{X}_s'$.

Hence, setting $\mathbf{W} = \mathbf{X}_s \backslash \mathbf{X}_s'$ to the values $\mathbf{w} = \mathbb{E}[\mathbf{W}|\mathrm{do}(\mathbf{X}_s' = \mathbf{x}_s[\mathbf{X}_s'])] + 1$ yields $\mathbb{E}[Y_i|\mathrm{do}(\mathbf{W} = \mathbf{w}, \mathbf{X}_s' = \mathbf{x}_s[\mathbf{X}_s'])] > \mathbb{E}[Y_i|\mathrm{do}(\mathbf{X}_s' = \mathbf{x}_s[\mathbf{X}_s'])]$, contradicting the assumption.

(Only if) Assume that $\mathbf{X}_s \not\subseteq \mathrm{an}(\mathbf{Y})_{\mathcal{G}_{\overline{\mathbf{X}}_s}}$. Then, for $\mathbf{X}_s' = \mathbf{X}_s \cap \mathrm{an}(\mathbf{Y})_{\mathcal{G}_{\overline{\mathbf{X}}_s}}$ it holds $\mathbf{X}_s' \subset \mathbf{X}_s$ and by the third rule of do-calculus, for every $\mathbf{x}_s \in \mathcal{D}(\mathbf{X}_s)$ it holds $\mathbb{E}[Y_i|\mathrm{do}(\mathbf{X}_s = \mathbf{x}_s)] = \mathbb{E}[Y_i|\mathrm{do}(\mathbf{X}_s' = \mathbf{x}_s[\mathbf{X}_s'])]$, $1 \leq i \leq m$. This is a contradiction because $\mathbf{X}_s$ was assumed to be a minimal intervention set. $\square$

## B.2. Partial-Orders among Intervention Sets

Recall the definition of possibly Pareto-optimal minimal intervention sets.

**Definition 4.3** (Possibly Pareto-optimal minimal intervention set). A set $\mathbf{X}_s \in \mathbb{M}_{\mathcal{G},\mathbf{Y}}$ is called *possibly Pareto-optimal* if, for at least one SCM conforming to $\mathcal{G}$, there exists $\mathbf{x}_s \in \mathcal{D}(\mathbf{X}_s)$ such that $(\mathbf{X}_s, \mathbf{x}_s)$ is Pareto-optimal for $\mathbb{P}(\mathbf{X})$, and for no $\mathbf{X}_s' \in \mathbb{M}_{\mathcal{G},\mathbf{Y}}\backslash\mathbf{X}_s, \mathbf{x}_s' \in \mathcal{D}(\mathbf{X}_s')$ it holds $\mu(\mathbf{X}_s', \mathbf{x}_s') \leq \mu(\mathbf{X}_s, \mathbf{x}_s)$, for all $1 \leq i \leq m$.

Characterizing such sets is the aim of this section. For simplicity, we first consider the special case in which $\mathcal{G}$ exhibits no unobserved confounders between $Y_i$ and any of its ancestors.

**Proposition 4.4.** *If no $Y_i$ is confounded with $\mathrm{an}(Y_i)_{\mathcal{G}}$ via unobserved confounders, then $\mathrm{pa}(\mathbf{Y})_{\mathcal{G}}$ is the only possibly Pareto-optimal minimal intervention set.*

*Proof.* Let $\mathbf{X}_s = \mathrm{pa}(\mathbf{Y})_{\mathcal{G}}$, and let $\mathrm{pa}(\mathbf{Y})_{\mathcal{G}} \neq \mathbf{X}_s'$ be another minimal intervention set with $\mathbf{x}_s' \in \mathcal{D}(\mathbf{X}_s')$. Define $\mathbf{Z} = \mathbf{X}_s\backslash(\mathbf{X}_s' \cap \mathbf{X}_s)$ and $\mathbf{W} = \mathbf{X}_s'\backslash(\mathbf{X}_s' \cap \mathbf{X}_s)$. Moreover, we choose an intervention value $\mathbf{x}_s^* \in \mathcal{D}(\mathbf{X}_s)$ such that it dominates $\mathbf{x}_s \in \mathcal{D}(\mathbf{X}_s)$ which is given by $\mathbf{x}_s[\mathbf{X}_s'] = \mathbf{x}_s'[\mathbf{X}_s]$ and $\mathbf{x}_s[\mathbf{Z}] = \mathbb{E}[\mathbf{Z}|\mathrm{do}(\mathbf{X}_s' = \mathbf{x}_s')]$. If $\mathbf{x}_s$ is non-dominated, define $\mathbf{x}_s^* = \mathbf{x}_s$. Then, for all $i = 1, \ldots, m$ it holds

$$\mathbb{E}[Y_i|\mathrm{do}(\mathbf{X}_s = \mathbf{x}_s^*)] = \mathbb{E}[Y_i|\mathrm{do}(\mathbf{X}_s \cap \mathbf{X}_s' = \mathbf{x}_s^*[\mathbf{X}_s'], \mathbf{Z} = \mathbf{x}_s^*[\mathbf{Z}])] \tag{8}$$

$$\leq \mathbb{E}[Y_i|\mathrm{do}(\mathbf{X}_s \cap \mathbf{X}_s' = \mathbf{x}_s[\mathbf{X}_s'], \mathbf{Z} = \mathbf{x}_s[\mathbf{Z}])] \tag{9}$$

$$= \mathbb{E}[Y_i|\mathrm{do}(\mathbf{X}_s \cap \mathbf{X}_s' = \mathbf{x}_s[\mathbf{X}_s'], \mathbf{Z} = \mathbf{x}_s[\mathbf{Z}], \mathbf{W} = \mathbf{x}_s'[\mathbf{W}])] \tag{10}$$

$$= \mathbb{E}[Y_i|\mathrm{do}(\mathbf{X}_s \cap \mathbf{X}_s' = \mathbf{x}_s[\mathbf{X}_s'], \mathbf{W} = \mathbf{x}_s'[\mathbf{W}]), \mathbf{Z} = \mathbf{x}_s[\mathbf{Z}]] \tag{11}$$

$$= \mathbb{E}[Y_i|\mathrm{do}(\mathbf{X}_s \cap \mathbf{X}_s' = \mathbf{x}_s[\mathbf{X}_s'], \mathbf{W} = \mathbf{x}_s'[\mathbf{W}])] \tag{12}$$

$$= \mathbb{E}[Y_i|\mathrm{do}(\mathbf{X}_s' = \mathbf{x}_s')], \tag{13}$$

where the inequality holds because $\mathbf{x}_s$ (weakly) dominates $\mathbf{x}_s^*$. Note that the second and third equalities are derived through the third and second rules of do-calculus, respectively. The second rule of do-calculus assumes that $Y_i$ is not confounded with $\mathrm{an}(Y_i)_{\mathcal{G}}$ via unobserved confounders. For $\mathbf{X}_s = \mathrm{pa}(\mathbf{Y})_{\mathcal{G}}$, it is possible to construct an SCM, conforming to $\mathcal{G}$, such that strict inequality holds for some $Y_i$, see the proof of Theorem B.2. This shows that $\mathrm{pa}(\mathbf{Y})_{\mathcal{G}}$ is the only possibly Pareto-optimal minimal intervention set. $\square$

We continue and study the more general case where unobserved confounders can be present between $Y_i$ and any of its ancestors. For this intent, we extend two existing concepts, called *minimal unobserved-confounders' territory* and *interventional border* (Lee & Bareinboim, 2018), to the multi-objective setting. Using these notions, we derive results which can fully characterize possibly Pareto-optimal minimal intervention sets in the aforementioned scenario.

**Definition 4.5** (Minimal unobserved confounders' territory). Let $\mathcal{H} = \mathcal{G}[\mathrm{An}(\mathbf{Y})_{\mathcal{G}}]$. A set of variables $\mathbf{T}$ in $\mathcal{H}$, with $\mathbf{Y} \subseteq \mathbf{T}$, is called a *UC-territory* for $\mathcal{G}$ w.r.t. $\mathbf{Y}$ if $\mathrm{De}(\mathbf{T})_{\mathcal{H}} = \mathbf{T}$ and $\mathrm{CC}(\mathbf{T})_{\mathcal{H}} = \mathbf{T}$. The UC-territory $\mathbf{T}$ is said to be *minimal*, denoted $\mathbf{T} = \mathrm{MUCT}(\mathcal{G}, \mathbf{Y})$, if no $\mathbf{T}' \subset \mathbf{T}$ is a UC-territory.

A minimal UC-territory for $\mathcal{G}$ w.r.t. $\mathbf{Y}$ can be constructed by extending a set of variables, starting from $\mathbf{Y}$, and iteratively updating the set with the c-component and descendants of the set. More intuitively, it is the minimal subset of $\mathrm{An}(\mathbf{Y})_{\mathcal{G}}$ that is governed by unobserved confounders, where at least one target $Y_i$ is adjacent to an unobserved confounder.

**Definition 4.6** (Interventional border). Let $\mathbf{T} = \mathrm{MUCT}(\mathcal{G}, \mathbf{Y})$. Then, $\mathbf{B} = \mathrm{pa}(\mathbf{T})_{\mathcal{G}}\backslash\mathbf{T}$ is called the *interventional border* for $\mathcal{G}$ w.r.t. $\mathbf{Y}$, denoted as $\mathrm{IB}(\mathcal{G}, \mathbf{Y})$.

We have already described these concepts in the main part. Before connecting the notion of minimal UC-territory and interventional border to possibly Pareto-optimal minimal intervention sets, we require the following proposition:

**Proposition B.1.** *Let* $\mathbf{T}$ *be a minimal UC-territory and* $\mathbf{B}$ *an interventional border for* $\mathcal{G}$ *w.r.t.* $\mathbf{Y}$. *Let* $\mathbf{X}_s \subseteq \mathbf{X}$ *be an intervention set and* $\mathbf{S} = (\mathbf{T} \cap \mathbf{X}_s) \cup \mathbf{B}$. *Then, for any* $\mathbf{x}_s \in \mathcal{D}(\mathbf{X}_s)$ *there exists* $\mathbf{s} \in \mathcal{D}(\mathbf{S})$ *such that* $\mathbb{E}[Y_i|\mathrm{do}(\mathbf{S} = \mathbf{s})] \leq \mathbb{E}[Y_i|\mathrm{do}(\mathbf{X}_s = \mathbf{x}_s)]$, *for all* $i = 1, \ldots, m$.

*Proof.* (Case $\mathbf{B} \subseteq \mathbf{X}_s$) Let $\mathbf{x}_s \in \mathcal{D}(\mathbf{X}_s)$ be an intervention value. Then, by the third rule of do-calculus, it holds $\mathbb{E}[Y_i|\mathrm{do}(\mathbf{X}_s = \mathbf{x}_s)] = \mathbb{E}[Y_i|\mathrm{do}(\mathbf{X}_s \cap (\mathbf{T} \cup \mathbf{B}) = \mathbf{x}_s[\mathbf{T} \cup \mathbf{B}])]$, $1 \leq i \leq m$. Since $\mathbf{X}_s \cap (\mathbf{T} \cup \mathbf{B}) = \mathbf{S}$, by setting $\mathbf{s} = \mathbf{x}_s[\mathbf{T} \cup \mathbf{B}]$, it follows $\mathbb{E}[Y_i|\mathrm{do}(\mathbf{X}_s = \mathbf{x}_s)] = \mathbb{E}[Y_i|\mathrm{do}(\mathbf{S} = \mathbf{s})]$.

(Case $\mathbf{B} \not\subseteq \mathbf{X}_s$) Let $\mathbf{x}_s \in \mathcal{D}(\mathbf{X}_s)$ be an intervention value. We define $\mathbf{B}' = \mathbf{S} \backslash (\mathbf{X}_s \cap \mathbf{S}) = \mathbf{S} \backslash (\mathbf{X}_s \cap (\mathbf{T} \cup \mathbf{B})) = \mathbf{B} \backslash (\mathbf{X}_s \cap \mathbf{B})$ and $\mathbf{W} = \mathbf{X}_s \backslash (\mathbf{X}_s \cap \mathbf{S}) = \mathbf{X}_s \backslash (\mathbf{X}_s \cap (\mathbf{T} \cup \mathbf{B}))$. Moreover, let $\mathbf{s}^* \in \mathcal{D}(\mathbf{S})$ such that it dominates $\mathbf{s} \in \mathcal{D}(\mathbf{S})$, which is given by $\mathbf{s}[\mathbf{B}'] = \mathbb{E}[\mathbf{B}'|\mathrm{do}(\mathbf{X}_s = \mathbf{x}_s)]$ and $\mathbf{s}[\mathbf{X}_s] = \mathbf{x}_s[\mathbf{T} \cup \mathbf{B}]$. If $\mathbf{s}$ is non-dominated, we set $\mathbf{s}^* = \mathbf{s}$. Then, for all $i = 1, \ldots, m$ it holds

$$\mathbb{E}[Y_i|\mathrm{do}(\mathbf{S} = \mathbf{s}^*)] = \mathbb{E}[Y_i|\mathrm{do}(\mathbf{X}_s \cap (\mathbf{T} \cup \mathbf{B}) = \mathbf{s}^*[\mathbf{X}_s], \mathbf{B}' = \mathbf{s}^*[\mathbf{B}'])] \tag{14}$$
$$\leq \mathbb{E}[Y_i|\mathrm{do}(\mathbf{X}_s \cap (\mathbf{T} \cup \mathbf{B}) = \mathbf{s}[\mathbf{X}_s], \mathbf{B}' = \mathbf{s}[\mathbf{B}'])] \tag{15}$$
$$= \mathbb{E}[Y_i|\mathrm{do}(\mathbf{X}_s \cap (\mathbf{T} \cup \mathbf{B}) = \mathbf{s}[\mathbf{X}_s], \mathbf{B}' = \mathbf{s}[\mathbf{B}'], \mathbf{W} = \mathbf{x}_s[\mathbf{W}])] \tag{16}$$
$$= \mathbb{E}[Y_i|\mathrm{do}(\mathbf{X}_s \cap (\mathbf{T} \cup \mathbf{B}) = \mathbf{s}[\mathbf{X}_s], \mathbf{W} = \mathbf{x}_s[\mathbf{W}]), \mathbf{B}' = \mathbf{s}[\mathbf{B}']] \tag{17}$$
$$= \mathbb{E}[Y_i|\mathrm{do}(\mathbf{X}_s \cap (\mathbf{T} \cup \mathbf{B}) = \mathbf{s}[\mathbf{X}_s], \mathbf{W} = \mathbf{x}_s[\mathbf{W}])] \tag{18}$$
$$= \mathbb{E}[Y_i|\mathrm{do}(\mathbf{X}_s = \mathbf{x}_s)], \tag{19}$$

where the inequality holds because $\mathbf{s}$ is (weakly) dominated by $\mathbf{s}^*$. Furthermore, the second and third equalities are derived through the third and second rules of do-calculus, respectively. $\square$

The following proposition is a building block for characterizing possibly Pareto-optimal minimal intervention sets via interventional borders. The proof is similar to the one given by Lee & Bareinboim (2018)

**Proposition B.2.** $\mathrm{IB}(\mathcal{G}, \mathbf{Y})$ *is a possibly Pareto-optimal minimal intervention set.*

*Proof.* The intuition of this proof is to construct an SCM, conforming to $\mathcal{G}$, for which the single best strategy involves intervening on $\mathrm{IB}(\mathcal{G}, \mathbf{Y})$. Let $\mathbf{T}$ and $\mathbf{B}$ denote $\mathrm{MUCT}(\mathcal{G}, \mathbf{Y})$ and $\mathrm{IB}(\mathcal{G}, \mathbf{Y})$, respectively. Every exogenous variable in $\mathbf{U}$ shall be a binary variable with its domain being $\{0, 1\}$. Let $\oplus$ denote the exclusive-or function and $\bigvee$ the logical OR operator.

(Case $\mathbf{T} = \mathbf{Y}$) In this case, $\mathbf{B}$ corresponds to the parents of $\mathbf{Y}$. Therefore, no target variable $Y_i$ is confounded with $\mathrm{an}(Y_i)_{\mathcal{G}}$ via unobserved confounders. Define an SCM such that

- Each endogenous variable $V \in \mathbf{V}$ is influenced by an exogenous variable $U_V \in \mathbf{V}$;

- $f_{Y_i} = \bigvee \mathbf{u}^{Y_i} \oplus \bigvee \mathbf{pa}_{Y_i}$ with $P(\mathbf{U}^{Y_i} = 0) \approx 1$, for all $i = 1, \ldots, m$;

- $f_X = (\bigoplus \mathbf{u}^X) \oplus (\bigoplus \mathbf{pa}_X)$ for $X \in \mathbf{X}$ and $P(U = 0) = 0.5$ for every $U \in \mathbf{U} \backslash (\bigcup_{i=1}^m \mathbf{U}^{Y_i})$.

By the third rule of do-calculus and by taking conditional expectations, it holds

$$\mathbb{E}[Y_i|\mathrm{do}(\mathbf{B} = 0)] = \mathbb{E}[Y_i|\mathrm{do}(\mathrm{pa}(Y_i)_{\mathcal{G}} = 0)] \tag{20}$$
$$= \mathbb{E}[Y_i|\mathrm{do}(\mathrm{pa}(Y_i)_{\mathcal{G}} = 0), \mathbf{U}^{Y_i} \neq 0]P(\mathbf{U}^{Y_i} \neq 0) + \mathbb{E}[Y_i|\mathrm{do}(\mathrm{pa}(Y_i)_{\mathcal{G}} = 0), \mathbf{U}^{Y_i} = 0]P(\mathbf{U}^{Y_i} = 0) \tag{21}$$
$$\approx 0 \tag{22}$$

for every $1 \leq i \leq m$. Meanwhile, all other interventions yield expectations greater than or equal to 0.5 in at least one component. Therefore, $\mathbf{B}$ is a possibly Pareto-optimal minimal intervention set.

(Case $\mathbf{T} \subset \mathbf{Y}$) In this case, at least one target variable $Y_i$ has an unobserved confounder with its ancestors. As a first step, it will be shown that there exists an SCM, conforming to $\mathcal{H} = \mathcal{G}[\mathbf{T} \cup \mathbf{B}]$, where the intervention $\mathrm{do}(\mathbf{B} = 0)$ is the single best strategy. To achieve this, we first define individual SCMs for each unobserved confounder in $\mathcal{H}[\mathbf{T}]$, and merge them into a single SCM where $\mathrm{do}(\mathbf{B} = 0)$ is indeed the best strategy. Let $\mathbf{U}' = \{U_j\}_{j=1}^k$ be the set of unobserved confounders in $\mathcal{H}[\mathbf{T}]$.

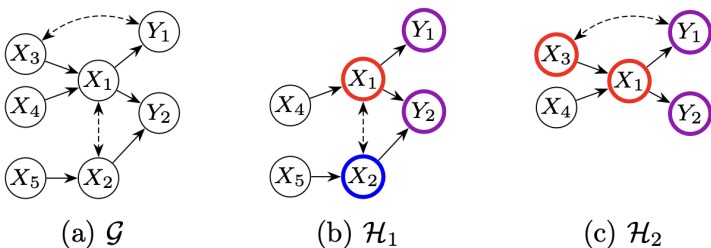

(a) $\mathcal{G}$        (b) $\mathcal{H}_1$        (c) $\mathcal{H}_2$

*Figure 8.* Original causal graph $\mathcal{G}$ and its color-coded subgraphs for each unobserved confounder.

*Table 3.* Values for $\mathcal{M}_1$, $\mathcal{M}_2$ and $\mathcal{M}$ given $X_4 = X_5 = 0$. The target variables are shown as bit sequences, $Y_1'$ and $Y_2'$, as well as binary values, $Y_1$ and $Y_2$.

| | | $\mathcal{M}_1$ | | | | $\mathcal{M}_2$ | | | | $\mathcal{M}$ | | | |
| $U_1$ | $U_2$ | $X_2^{(1)}$ | $X_1^{(1)}$ | $Y_1^{(1)}$ | $Y_2^{(1)}$ | $X_3^{(2)}$ | $X_1^{(2)}$ | $Y_1^{(2)}$ | $Y_2^{(2)}$ | $Y_1'$ | $Y_2'$ | $Y_1$ | $Y_2$ |
|---|---|---|---|---|---|---|---|---|---|---|---|---|---|
| 0 | 0 | 0 | 1 | 2 | 2 | 1 | 1 | 2 | 2 | 1010 | 1010 | 0 | 0 |
| 0 | 1 | | | | | 0 | 0 | 1 | 1 | 0110 | 0110 | 0 | 0 |
| 1 | 0 | 1 | 0 | 1 | 1 | 1 | 1 | 2 | 2 | 1001 | 1001 | 0 | 0 |
| 1 | 1 | | | | | 0 | 0 | 1 | 1 | 0101 | 0101 | 0 | 0 |

Given $U_j \in \mathbf{U}'$, let $B^{(j)}$ and $R^{(j)}$ denote its two children. We define an SCM $\mathcal{M}_j$, where the graph structure is given by

$$\mathcal{H}_j = \mathcal{H} \left[ \text{De} \left( \left\{ B^{(j)}, R^{(j)} \right\} \right)_{\mathcal{H}} \cup \left( \mathbf{B} \cap \text{pa} \left( \text{De} \left( \left\{ B^{(j)}, R^{(j)} \right\} \right)_{\mathcal{H}} \right) \right) \right], \tag{23}$$

and all bidirected edges, except for $U_j$, are removed. In order to set the structural equations for variables in $\mathcal{H}_j$, the vertices will be labelled via colour coding: Let vertices in $\text{De}\left(B^{(j)}\right)_{\mathcal{H}} \backslash \text{De}\left(R^{(j)}\right)_{\mathcal{H}}$ be labelled as blue, $\text{De}\left(R^{(j)}\right)_{\mathcal{H}} \backslash \text{De}\left(B^{(j)}\right)_{\mathcal{H}}$ as red, and $\text{De}\left(B^{(j)}\right)_{\mathcal{H}} \cap \text{De}\left(R^{(j)}\right)_{\mathcal{H}}$ as purple. All target variables are coloured as purple as well. Moreover, $B^{(j)}$ and $R^{(j)}$ shall perceive $U_j$ as a parent coloured as blue with value $U_j$ and red with value $1 - U_j$, respectively. The blue-, red- and purple-coloured variables are set to 3 if any of their parents in $\mathbf{B}$ is not 0. Otherwise, their values are determined as follows. For every blue and red vertex, the associated structural equation returns the common value of its parents of the same colour and returns 3 if coloured parents' values are not homogeneous. For every purple vertex, its corresponding equation returns 2 if every blue, red and purple parent is 0,1, and 2, respectively, and returns 1 if 1,0,1, respectively.

Next, the SCMs $\mathcal{M}_1, \ldots, \mathcal{M}_k$ will be merged into one single SCM, that conforms to $\mathcal{H}$, and for which $\text{do}(\mathbf{B} = 0)$ is the single best intervention. Note that in $\mathcal{M}_j$ all variables can be represented with just two bits. To construct a unified SCM, variables in $\mathbf{T}$ are represented with $2k$ bits, where $\mathcal{M}_j$ takes the $2j-1^{\text{th}}$ and $2j^{\text{th}}$ bits. Every target variable $Y_i$ is represented as a sequence of bits and binarised as follows. $Y_i$ is set to 0 if its $2j-1^{\text{th}}$ and $2j^{\text{th}}$ bits are 00, 01 or 10 for every $1 \leq j \leq k$, and 1 otherwise. Let $P(U_j = 1) = 0.5$ for $U_j \in \mathbf{U}'$. Therefore, it holds $Y_i = 0$ if $\text{do}(\mathbf{B} = 0)$ and $Y_i = 1$ if $\text{do}(\mathbf{B} \neq 0)$. If any variable in $\mathbf{T}$ is intervened, then at least one SCM $\mathcal{M}_j$ will be disrupted, resulting in an expectation larger than or equal to 0.5 for at least one target variable. In the multi-target setting, it may happen that some target variables do not occur in any of the $\mathcal{M}_j$'s. This happens if a target $Y_i$ has no parents in $\mathbf{T}$, but only in $\mathbf{B}$. For all such $Y_i$'s, we set $f_{Y_i} = \mathbf{u}^{Y_i} \oplus \bigvee \mathbf{pa}_{Y_i}$ with $P(\mathbf{U}^{Y_i} = 0) \approx 1$. As such, the newly constructed SCM enforces $\mathbb{E}[Y_i | \text{do}(\mathbf{B} = 0)] \approx 0$. Meanwhile, all other interventions yield expectations greater than or equal to 0.5

As a last step, the previously defined SCM for $\mathcal{H} = \mathcal{G}[\mathbf{T} \cup \mathbf{B}]$, will be extended to an SCM for $\mathcal{G}$. However, we can ignore joint probability distributions for any exogenous variables only affecting endogenous variables outside of $\mathcal{H}$. Setting structural equations for endogenous variables outside of $\mathcal{H}$ is redundant as well. For $V \in \text{An}(\mathbf{Y})_{\mathcal{G}} \backslash \mathbf{T}$, we define the structural equations as $f_V = (\bigoplus \mathbf{u}^V) \oplus (\bigoplus \mathbf{pa}_V)$. For $U \in \mathbf{U} \backslash \mathbf{U}'$, we set $P(U = 0) = 0.5$ if $U$'s child(ren) is disjoint to $\mathbf{T}$, and $P(U = 0) \approx 1$ otherwise. Note that $\text{do}(\mathbf{B} = 0)$ is still the single optimal intervention. Therefore, $\mathbf{B}$ is a possibly

Pareto-optimal minimal intervention set. □

In order to illustrate the construction of an SCM where $\text{do}(\text{IB}(\mathcal{G}, \mathbf{Y}) = 0)$ is the single best strategy, consider Figure 8, showing an exemplary graph and its colour-coded subgraphs, $\mathcal{H}_1$ and $\mathcal{H}_2$, for each unobserved confounder. Table 3 presents the associated values for $\mathcal{M}_1$ and $\mathcal{M}_2$, as well as values for the target variables in the final SCM $\mathcal{M}$. The next proposition generalizes the previous one.

**Proposition 4.7.** $\text{IB}(\mathcal{G}_{\overline{\mathbf{X}}_s}, \mathbf{Y})$ *is a possibly Pareto-optimal minimal intervention set for any* $\mathbf{X}_s \in \mathbb{P}(\mathbf{X})$.

*Proof.* Let $\mathbf{X}_s$ be an intervention set. Let us denote $\mathbf{T} = \text{MUCT}(\mathcal{G}_{\overline{\mathbf{X}}_s}, \mathbf{Y})$, $\mathbf{B} = \text{IB}(\mathcal{G}_{\overline{\mathbf{X}}_s}, \mathbf{Y})$ and $\mathbf{T}_0 = \text{MUCT}(\mathcal{G}, \mathbf{Y})$. Using the strategy from Theorem 4.7, we construct an SCM for $\mathcal{G}[\mathbf{T} \cup \mathbf{B}]$ while ignoring unobserved confounders between $\mathbf{T}$ and $\mathbf{T}_0 \backslash \mathbf{T}$. Let $\mathbf{U}'$ be the set of such unobserved confounders. Now, the SCM needs to be modified to ensure that $\text{do}(\mathbf{B} = 0)$ is the single best intervention. Every $U \in \mathbf{U}'$ shall flip (i.e., $0 \leftrightarrow 1$) the value of its endogenous child in $\mathbf{T}$ whenever $U = 1$. Let $P(U = 0) \approx 1$, so that it holds $\mathbb{E}[Y_i | \text{do}(\mathbf{B} = 0)] \approx 0$. Intervening on $\mathbf{B} \neq 0$ or on any variable in $\mathbf{T}$ results in expectations around 0.5 or above. □

Notably, Proposition 4.7 extends Proposition B.2 when $\mathbf{X}_s \neq \varnothing$. Note that, by iterating over all intervention sets $\mathbf{X}_s \in \mathbb{P}(\mathbf{X})$, we can discover possibly Pareto-optimal minimal intervention sets in a given graph. The following theorem is an extension of the main result by Lee & Bareinboim (2018) to the scenario where multiple target variables are present. It shows that the aforementioned strategy suffices to find not some, but all, such sets.

**Theorem 4.8.** *A set* $\mathbf{X}_s$ *is a possibly Pareto-optimal minimal intervention set if and only if it holds* $\text{IB}(\mathcal{G}_{\overline{\mathbf{X}}_s}, \mathbf{Y}) = \mathbf{X}_s$.

*Proof.* (If) This is a special case of Proposition 4.7.

(Only if) Let $\mathbf{X}_s$ be a minimal intervention set and $\mathbf{x}_s \in \mathcal{D}(\mathbf{X}_s)$ an intervention value. Denote $\mathbf{T} = \text{MUCT}(\mathcal{G}_{\overline{\mathbf{X}}_s}, \mathbf{Y})$, $\mathbf{B} = \text{IB}(\mathcal{G}_{\overline{\mathbf{X}}_s}, \mathbf{Y})$, $\mathbf{T}_0 = \text{MUCT}(\mathcal{G}, \mathbf{Y})$ and $\mathbf{B}_0 = \text{IB}(\mathcal{G}, \mathbf{Y})$. From Theorem B.1, we know that no POMIS intersects with $\text{An}(\mathbf{B}_0)_{\mathcal{G}} \backslash \mathbf{B}_0$ and thus, it is possible to conclude $\mathbf{X}_s \subseteq \mathbf{T}_0 \cup \mathbf{B}_0 \backslash \mathbf{Y}$. Note that it holds $\mathbf{X}_s \subseteq \text{An}(\mathbf{B})_{\mathcal{G}}$ since otherwise it would follow $\mathbf{X}_s \cap \mathbf{T} \neq \emptyset$, which contradicts that $\mathbf{X}_s$ is neither a descendant of some variable nor confounded in $\mathcal{G}_{\overline{\mathbf{X}}_s}$. Let $\mathbf{B}' = \mathbf{B} \backslash (\mathbf{X}_s \cap \mathbf{B})$ and $\mathbf{W} = \mathbf{X}_s \backslash (\mathbf{X}_s \cap \mathbf{B})$. Moreover, we define an intervention value $\mathbf{b}^* \in \mathcal{D}(\mathbf{B})$ such that it dominates $\mathbf{b} \in \mathcal{D}(\mathbf{B})$, which is given by $\mathbf{b}[\mathbf{B}'] = \mathbb{E}[\mathbf{B}' | \text{do}(\mathbf{X}_s = \mathbf{x}_s)]$ and $\mathbf{b}[\mathbf{X}_s] = \mathbf{x}_s[\mathbf{B}]$. If $\mathbf{b}$ is non-dominated, we set $\mathbf{b}^* = \mathbf{b}$. Then, for all $i = 1, \ldots, m$, it holds

$$\mathbb{E}[Y_i | \text{do}(\mathbf{B} = \mathbf{b}^*)] = \mathbb{E}[Y_i | \text{do}(\mathbf{B} \cap \mathbf{X}_s = \mathbf{b}^*[\mathbf{X}_s], \mathbf{B}' = \mathbf{b}^*[\mathbf{B}'])] \tag{24}$$
$$\geq \mathbb{E}[Y_i | \text{do}(\mathbf{B} \cap \mathbf{X}_s = \mathbf{b}[\mathbf{X}_s], \mathbf{B}' = \mathbf{b}[\mathbf{B}'])] \tag{25}$$
$$= \mathbb{E}[Y_i | \text{do}(\mathbf{B} \cap \mathbf{X}_s = \mathbf{b}[\mathbf{X}_s], \mathbf{B}' = \mathbf{b}[\mathbf{B}'], \mathbf{W} = \mathbf{x}_s[\mathbf{W}])] \tag{26}$$
$$= \mathbb{E}[Y_i | \text{do}(\mathbf{B} \cap \mathbf{X}_s = \mathbf{b}[\mathbf{X}_s], \mathbf{W} = \mathbf{x}_s[\mathbf{W}]), \mathbf{B}' = \mathbf{b}[\mathbf{B}']] \tag{27}$$
$$= \mathbb{E}[Y_i | \text{do}(\mathbf{B} \cap \mathbf{X}_s = \mathbf{b}[\mathbf{X}_s], \mathbf{W} = \mathbf{x}_s[\mathbf{W}])] \tag{28}$$
$$= \mathbb{E}[Y_i | \text{do}(\mathbf{X}_s = \mathbf{x}_s)], \tag{29}$$

where the inequality holds because $\mathbf{b}$ is (weakly) dominated by $\mathbf{b}^*$. Furthermore, the second and third equalities are derived through the third and second rules of do-calculus, repectively. □

Theorem 4.8 provides a necessary and sufficient condition for a set of variables to be a possibly Pareto-optimal minimal intervention set. The proof of the theorem gives the following corollary:

**Corollary 4.9.** *Let* $\mathbf{X}_s \in \mathbb{P}(\mathbf{X})$ *and* $\mathbf{X}'_s = \text{IB}(\mathcal{G}_{\overline{\mathbf{X}}_s}, \mathbf{Y})$. *For any* $\mathbf{x}_s \in \mathcal{D}(\mathbf{X}_s)$ *there exist* $\mathbf{x}'_s \in \mathcal{D}(\mathbf{X}'_s)$ *such that it holds* $\mu(\mathbf{X}'_s, \mathbf{x}'_s) \leq \mu(\mathbf{X}_s, \mathbf{x}_s)$, *for all* $1 \leq i \leq m$.

## C. MO-CBO Problems

In this section, we present the collection of synthetic and real-world causal graphs used to evaluate the performance of MO-CBO in comparison to standard MOBO baselines.

**Cost of Interventions**   The MO-CBO algorithm requires evaluating the objective functions which inherently involves implementing interventions on the system. However, in practical scenarios, such interventions can be costly, making it important to prioritize sample efficiency and explicitly account for the cost of an intervention $do(\mathbf{X}_s = \mathbf{x}_s)$, denoted as $\text{cost}(\mathbf{X}_s, \mathbf{x}_s)$. We use the convention $\text{cost}(\mathbf{X}_s, \mathbf{x}_s) = \sum_{X_i \in \mathbf{X}_s, x_i = \mathbf{x}_s[X_i]} \text{cost}(X_i, x_i)$. Each experiment is conducted under a fixed cost budget, terminating once the budget is exhausted. To reflect practical constraints, every MO-CBO scenario includes a tailored cost structure that accurately represents the varying difficulty or expense of different interventions.

### C.1. SYNTHETIC-1

We introduce the first synthetic MO-CBO problem in our experimental study, referred to as SYNTHETIC-1, which is defined by the causal graph $\mathcal{G}$ and associated structural assignments presented in Figure 9. The interventional domains are specified as $\mathcal{D}(X_1), \mathcal{D}(X_2) = [-1, 2]$ and $\mathcal{D}(X_3), \mathcal{D}(X_4) = [-1, 1]$. Moreover, all exogenous variables follow the standard normal distribution, and there are no unobserved confounders. All treatment variables $X_i, 1 \leq i \leq 4$, shall have fixed unit cost of $\text{cost}(X_i, x_i) = 1$ for all $x_i \in \mathcal{D}(X_i)$.

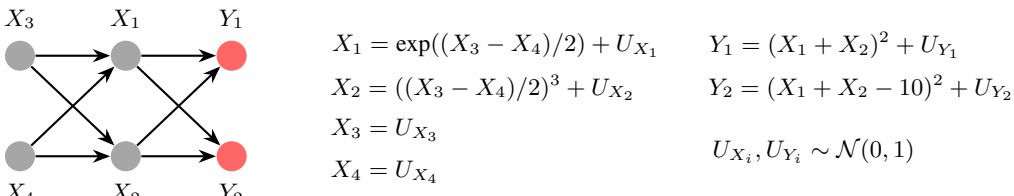

$$X_1 = \exp((X_3 - X_4)/2) + U_{X_1} \qquad Y_1 = (X_1 + X_2)^2 + U_{Y_1}$$
$$X_2 = ((X_3 - X_4)/2)^3 + U_{X_2} \qquad Y_2 = (X_1 + X_2 - 10)^2 + U_{Y_2}$$
$$X_3 = U_{X_3}$$
$$X_4 = U_{X_4} \qquad\qquad U_{X_i}, U_{Y_i} \sim \mathcal{N}(0, 1)$$

*Figure 9.* SYNTHETIC-1. An SCM consisting of four treatment and two output variables, depicted with grey and red nodes, respectively. There are no unobserved confounders.

### C.2. SYNTHETIC-2

SYNTHETIC-2 is the next MO-CBO problem of our experimental study, defined by the causal graph $\mathcal{G}$ and associated structural equations in Figure 10. The interventional domains are $\mathcal{D}(X_1) = [-2, 5]$, $\mathcal{D}(X_4) = [-4, 5]$ and $\mathcal{D}(X_i) = [0, 5]$ for $i = 1, 2$. Moreover, the exogenous variables $U_{X_i}, U_{Y_i}$ follow a Gaussian distribution, and there is an unobserved confounder $U$ influencing the target variable $Y_1$ and its ancestor $X_4$. All treatment variables $X_i, 1 \leq i \leq 4$, have fixed unit cost of $\text{cost}(X_i, x_i) = 1$ for all $x_i \in \mathcal{D}(X_i)$.

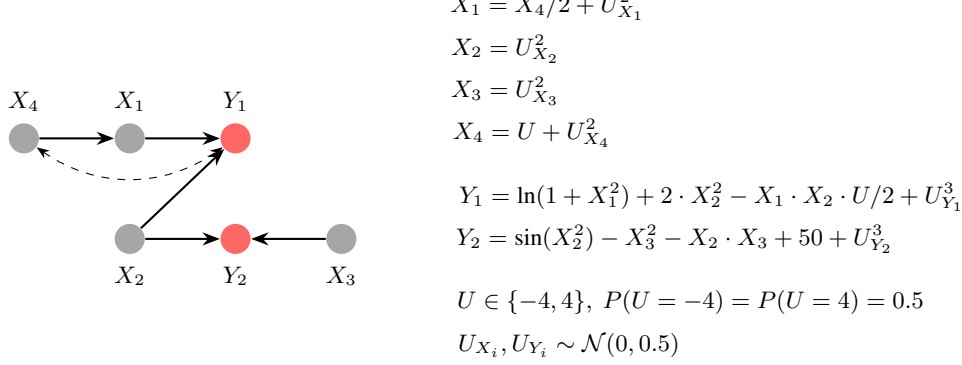

$$X_1 = X_4/2 + U_{X_1}^2$$
$$X_2 = U_{X_2}^2$$
$$X_3 = U_{X_3}^2$$
$$X_4 = U + U_{X_4}^2$$

$$Y_1 = \ln(1 + X_1^2) + 2 \cdot X_2^2 - X_1 \cdot X_2 \cdot U/2 + U_{Y_1}^3$$
$$Y_2 = \sin(X_2^2) - X_3^2 - X_2 \cdot X_3 + 50 + U_{Y_2}^3$$

$$U \in \{-4, 4\}, \ P(U = -4) = P(U = 4) = 0.5$$
$$U_{X_i}, U_{Y_i} \sim \mathcal{N}(0, 0.5)$$

*Figure 10.* SYNTHETIC-2. An SCM consisting of four treatment and two output variables, depicted with grey and red nodes, respectively. It includes an unobserved confounder, denoted via the dashed bi-directed edge, affecting one output and its ancestor.

## C.3. HEALTH

The MO-CBO problem HEALTH is defined by the causal graph and structural equations in Figure 11. This model originates from previous works of Ferro et al. (2015), and is based on real-world causal relationships. It captures prostate-specific-antigen (PSA) levels in causal relation to its risk factors, such as BMI, calorie intake (CI) and aspirin usage. The variable Aspirin indicates the daily aspirin regimen while Statin denotes a subject' statin medication. Additionally, PSA represents the total antigen level circulating in a subject's blood, measured in ng/mL. For patients sensitive to Statin medications, the aim is to determine how to manipulate relevant risk factors to minimize both Statin and PSA. To this end, we treat both Statin and PSA as target variables. The treatment variables include BMI, Weight, CI, and Aspirin usage with interventional domains $\mathcal{D}(\text{BMI}) = [20, 30]$, $\mathcal{D}(\text{Weight}) = [50, 100]$, $\mathcal{D}(\text{CI}) = [-100, 100]$ and $\mathcal{D}(\text{Aspirin}) = [0, 1]$. We choose to consider a specific age groups of interest, and define $U_{\text{age}}$ as a Gaussian random variable with mean 65 and standard deviation 1, focusing on individuals close to the age of 65. The variables CI and Aspirin are set to have fixed unit cost, i.e. $\text{cost}(X, x) = 1$ for $X \in \{\text{CI, Aspirin}\}$, $x \in \mathcal{D}(X)$. Since BMI and weight are significantly harder to treat, we increase their cost to $\text{cost}(X, x) = 3$ for $X \in \{\text{BMI, weight}\}$, $x \in \mathcal{D}(X)$.

The single-objective version of HEALTH, aiming to minimize only PSA, has previously been used to demonstrate the applicability of CBO (Aglietti et al., 2020), as well as for several of its variants (e.g. Gultchin et al. (2023) and Aglietti et al. (2023)).

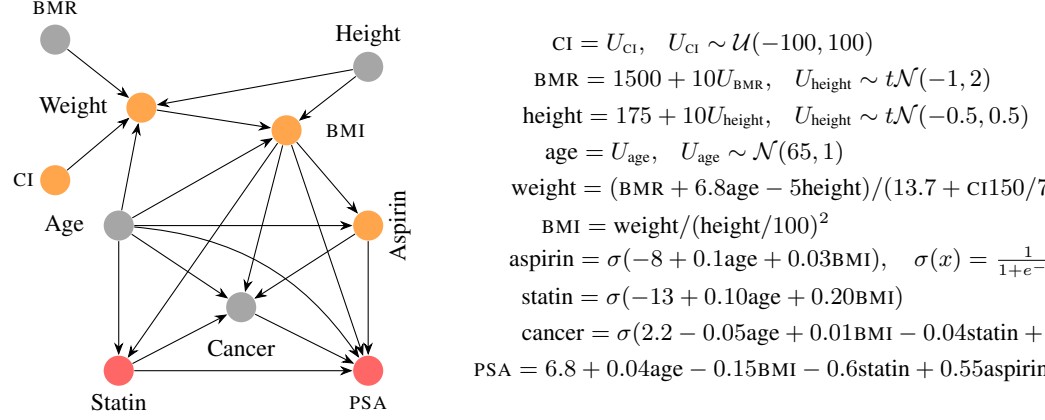

$$\text{CI} = U_{\text{CI}}, \quad U_{\text{CI}} \sim \mathcal{U}(-100, 100)$$
$$\text{BMR} = 1500 + 10 U_{\text{BMR}}, \quad U_{\text{height}} \sim t\mathcal{N}(-1, 2)$$
$$\text{height} = 175 + 10 U_{\text{height}}, \quad U_{\text{height}} \sim t\mathcal{N}(-0.5, 0.5)$$
$$\text{age} = U_{\text{age}}, \quad U_{\text{age}} \sim \mathcal{N}(65, 1)$$
$$\text{weight} = (\text{BMR} + 6.8\text{age} - 5\text{height})/(13.7 + \text{CI}150/7716)$$
$$\text{BMI} = \text{weight}/(\text{height}/100)^2$$
$$\text{aspirin} = \sigma(-8 + 0.1\text{age} + 0.03\text{BMI}), \quad \sigma(x) = \frac{1}{1+e^{-x}}$$
$$\text{statin} = \sigma(-13 + 0.10\text{age} + 0.20\text{BMI})$$
$$\text{cancer} = \sigma(2.2 - 0.05\text{age} + 0.01\text{BMI} - 0.04\text{statin} + 0.02\text{aspirin})$$
$$\text{PSA} = 6.8 + 0.04\text{age} - 0.15\text{BMI} - 0.6\text{statin} + 0.55\text{aspirin} + \text{cancer} + U_{\text{PSA}}$$
$$U_{\text{PSA}} \sim \mathcal{N}(0, 0.4)$$

*Figure 11.* HEALTH. An SCM with relations between variables such as age, BMI, aspirin and statin usage, and their effects on PSA levels (Gultchin et al., 2023). $\mathcal{U}(\cdot, \cdot)$ denotes a uniform distribution and $t\mathcal{N}(a, b)$ a standard Gaussian distribution truncated between $a$ and $b$. Red, orange, and grey nodes depict target, manipulative, and non-manipulative variables, respectively.

## C.4. CREDIT APPROVAL

The final MO-CBO problem in our experiments is called CREDIT APPROVAL, and is specified by the causal graph and structural equations shown in Figure 12. This problem models the probability of credit approval as a function of various demographic and financial variables, including age, gender, education, loan amount, loan duration, income, and savings. It is based on the German Credit UCI dataset (Murphy, 1994) and causal relationships adapted from in (Karimi et al., 2020). We treat both approval probability and loan duration as target variables. The treatment variables are education, loan amount, income and savings, with interventional domains given as: $\mathcal{D}(\text{education}) = [-0.5, 0.5]$, $\mathcal{D}(\text{loan amount}) = [-1, 2]$, $\mathcal{D}(\text{income}) = [-2, 1]$, and $\mathcal{D}(\text{savings}) = [-5, 1]$. Note that the variables age, education, loan amount, loan duration, income and savings are modelled as a deviations from their means. To reduce complexity and focus our analysis, we fix the gender variable to 0 (e.g., male), diverging from the original specification by Karimi et al. (2020), where gender is defined as Bernoulli(0.5). Additionally, we assume that all other variables remain close to their mean values and reduce the level of observational noise compared to Karimi et al. (2020). The variables loan amount and income are set to have fixed unit cost, i.e. $\text{cost}(X, x) = 1$ for $X \in \{\text{loan amount, income}\}$, $x \in \mathcal{D}(X)$. For education and savings, we increase the cost to $\text{cost}(X, x) = 2$ for $X \in \{\text{education, savings}\}$, $x \in \mathcal{D}(X)$.

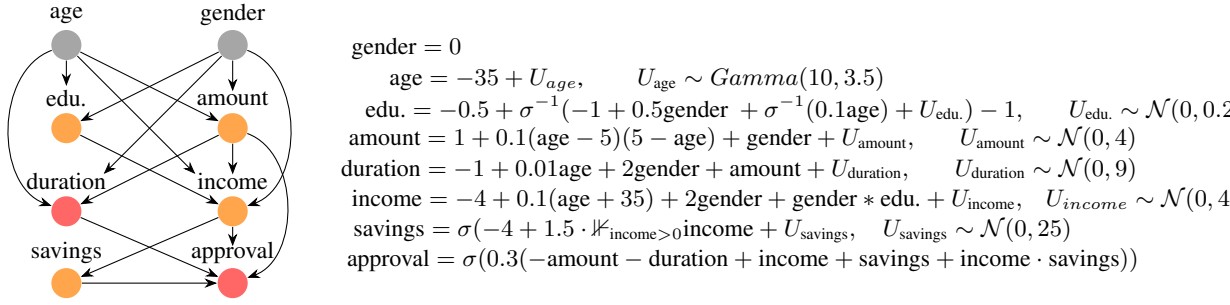

$$
\begin{aligned}
&\text{gender} = 0 \\
&\quad\text{age} = -35 + U_{age}, \qquad U_{\text{age}} \sim Gamma(10, 3.5) \\
&\quad\text{edu.} = -0.5 + \sigma^{-1}(-1 + 0.5\text{gender} + \sigma^{-1}(0.1\text{age}) + U_{\text{edu.}}) - 1, \qquad U_{\text{edu.}} \sim \mathcal{N}(0, 0.25) \\
&\quad\text{amount} = 1 + 0.1(\text{age} - 5)(5 - \text{age}) + \text{gender} + U_{\text{amount}}, \qquad U_{\text{amount}} \sim \mathcal{N}(0, 4) \\
&\quad\text{duration} = -1 + 0.01\text{age} + 2\text{gender} + \text{amount} + U_{\text{duration}}, \qquad U_{\text{duration}} \sim \mathcal{N}(0, 9) \\
&\quad\text{income} = -4 + 0.1(\text{age} + 35) + 2\text{gender} + \text{gender} * \text{edu.} + U_{\text{income}}, \quad U_{income} \sim \mathcal{N}(0, 4) \\
&\quad\text{savings} = \sigma(-4 + 1.5 \cdot \mathbb{1}_{\text{income}>0}\text{income} + U_{\text{savings}}, \quad U_{\text{savings}} \sim \mathcal{N}(0, 25) \\
&\quad\text{approval} = \sigma(0.3(-\text{amount} - \text{duration} + \text{income} + \text{savings} + \text{income} \cdot \text{savings}))
\end{aligned}
$$

*Figure 12.* CREDIT APPROVAL. An SCM which models the probability of credit approval as a function of various demographic and financial variables (Karimi et al., 2020). Red, orange, and grey nodes depict target, manipulative, and non-manipulative variables, respectively.

## D. Experiments

### D.1. Hyperparameters

**MO-CBO algorithm** The DGEMO backbone of our MO-CBO algorithm has mostly the same hyperparameters as its original implementation from Konakovic Lukovic et al. (2020). The batch size is set to 5.

**ParEGO** We adopt a batch variant of ParEGO by using $b$ random scalarization weights in each iteration, with $b$ being the batch size. Moreover, Chebyshev scalarization (Miettinen, 1999) and the CMA-ES algorithm (Hansen, 2023) are used to solve the scalarized single-objective problems with $\sigma = 0.5$ as initial standard deviation.

**MOEA/D-EGO** Following Konakovic Lukovic et al. (2020), our implementation of the MOEA/D-EGO baseline follows its original framework, with the key difference being the removal of FuzzyCM. Given the current computational efficiency of training Gaussian process models, they opt to use them directly for prediction instead of relying on faster but less accurate approximation techniques. As a result, this version may offer improved performance due to the enhanced predictive accuracy. We employ simulated binary crossover and polynomial mutation for MOEA/D, with the remaining hyperparameters detailed in Konakovic Lukovic et al. (2020).

**TSEMO** For TSEMO, we largely adopt the same hyperparameter settings as in the original implementation. Specifically, we use 100 points for spectral sampling.

*q***NEHVI** We implement qNEHVI using the botorch library. We use 10 optimization restarts, and 64 raw samples for acquisition maximization. Moreover, the acquisition function uses a Sobol QMC sampler with 128 samples.

**DGEMO** For DGEMO, we retain the hyperparameter configuration from its original implementation.

### D.2. Performance Metrics

We require metrics to assess the quality of a given Pareto front approximation. These metrics evaluate both the convergence of the approximated front to the true front and the diversity of the solutions across the performance space. For a given optimisation problem, let $\mathbf{A}$ be the set of points from an approximated Pareto front. If the ground-truth Pareto front is known, it is possible to evaluate how well $\mathbf{A}$ approximates it given the following two metrics. Let $\mathbf{Z}$ be the set of points on the true Pareto front.

**Generational distance (GD)** A common performance indicator for evaluating a given Pareto front approximation is the so-called *generational distance* (Schutze et al., 2012). It is the average distance from any point $\mathbf{a}_i \in \mathbf{A}$ to its closest point in the Pareto front $\mathbf{Z}$. Formally,

$$
\text{GD}(\mathbf{A}) = \left( \frac{1}{|\mathbf{A}|} \sum_{i=1}^{|\mathbf{A}|} d_i^p \right)^{1/p}, \tag{30}
$$

where $d_i^p$ is the Euclidean distance from $\mathbf{a}_i$ to its nearest point in $\mathbf{Z}$. We set $p = 2$ in our experiments.

**Inverted generational distance (IGD)**   The *inverted generational distance* measures the distance from any point $\mathbf{z}_i \in \mathbf{Z}$ to its closest point in $\mathbf{A}$ (Schutze et al., 2012). Thereby, it can serve as an indicator of the coverage given by the approximated front. Formally,

$$\text{IGD}(\mathbf{A}) = \left( \frac{1}{|\mathbf{Z}|} \sum_{i=1}^{|\mathbf{Z}|} \hat{d}_i^{\;p} \right)^{1/p}, \tag{31}$$

where $d_i^p$ is the Euclidean distance from $\mathbf{z}_i$ to its nearest point in $\mathbf{A}$. We set $p = 2$ in our experiments.

The GD evaluates the convergence of the approximated Pareto front to the true front, whereas the IGD measures the diversity of the solutions across the output space. For both metrics, smaller values indicate a better approximation of the true Pareto front.

### D.3. Runtime

All experiments were executed on a machine equipped with an Apple M2 processor and 8GB of RAM. The average runtimes are reported in Table 4.

Table 4. Algorithm runtime comparison (seconds per iteration), averaged across 10 seeds.

| Problem | ParEGO | MOEA/D-EGO | TSEMO | $q$NEHVI | DGEMO | MO-CBO (ours) |
|---|---|---|---|---|---|---|
| SYNTHETIC-1 | 14.39 | 0.62 | 0.20 | 14.23 | 4.35 | 3.43 |
| SYNTHETIC-2 | 9.29 | 0.74 | 0.35 | 12.09 | 4.47 | 3.82 |
| HEALTH | 8.65 | 4.19 | 3.81 | 18.98 | 6.26 | 6.12 |
| CREDIT APPROVAL | 10.13 | 0.77 | 0.34 | 20.87 | 26.45 | 10.55 |

## D.4. Results

**SYNTHETIC-1**   We present the experimental results for the MO-CBO problem SYNTHETIC-1, see Figure 13. We observe that our MO-CBO algorithm consistently outperforms all MOBO baselines, yielding more diverse and well-distributed solutions across the target Pareto front $\mathcal{P}_f^{\mathsf{c}}(\mathbb{P}(\mathbf{X}))$. All methods were run until a fixed cost budget of 150 was exhausted.

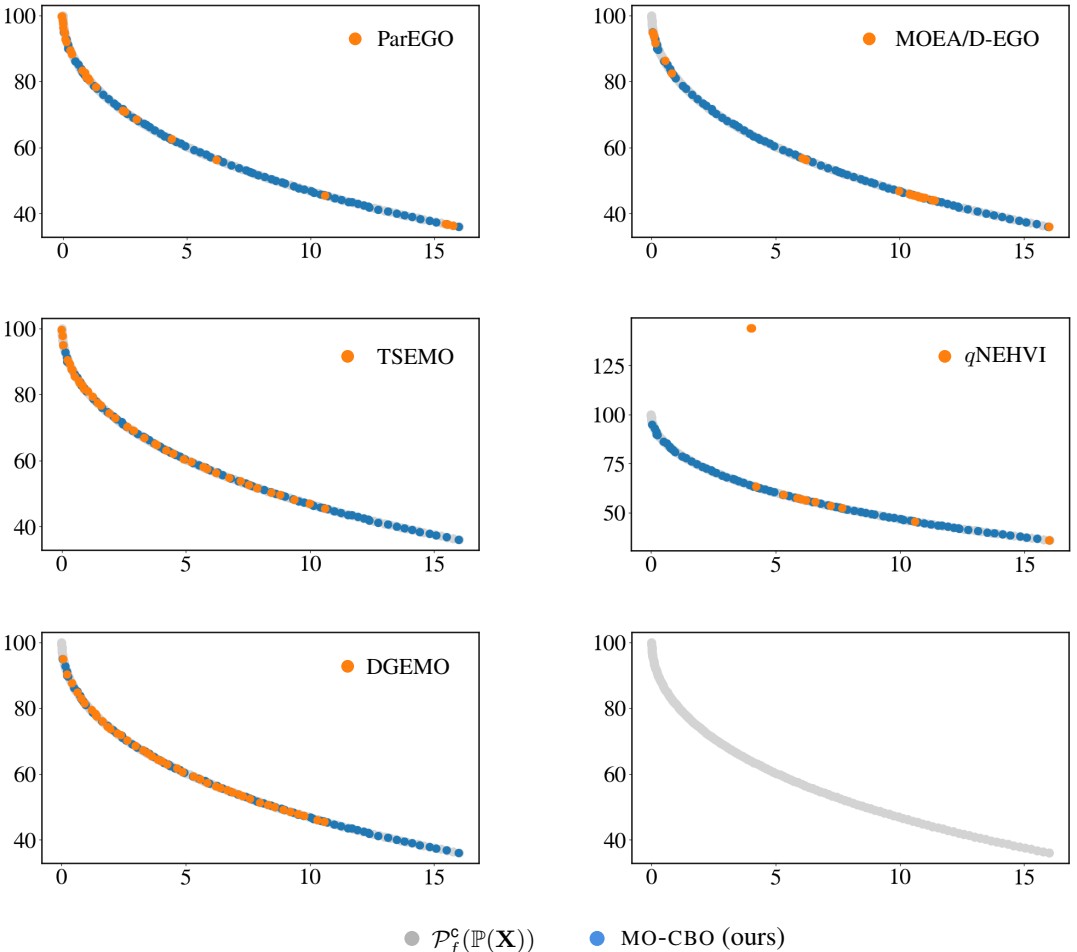

*Figure 13.* SYNTHETIC-1. Comparison of Pareto front approximations produced by our MO-CBO algorithm and various MOBO baselines: ParEGO, MOEA/D-EGO, TSEMO, $q$NEHVI, and DGEMO. The $x$-axis corresponds to objective $Y_1$, and the $y$-axis to $Y_2$.

**SYNTHETIC-2** We present the experimental results for the MO-CBO problem SYNTHETIC-2, see Figure 14. We observe that the solutions identified by our MO-CBO algorithm tightly fit the target Pareto front $\mathcal{P}_f^{\mathsf{c}}(\mathbb{P}(\mathbf{X}))$, and strictly dominate those found by the MOBO baselines. All methods were run until a fixed cost budget of 200 was exhausted.

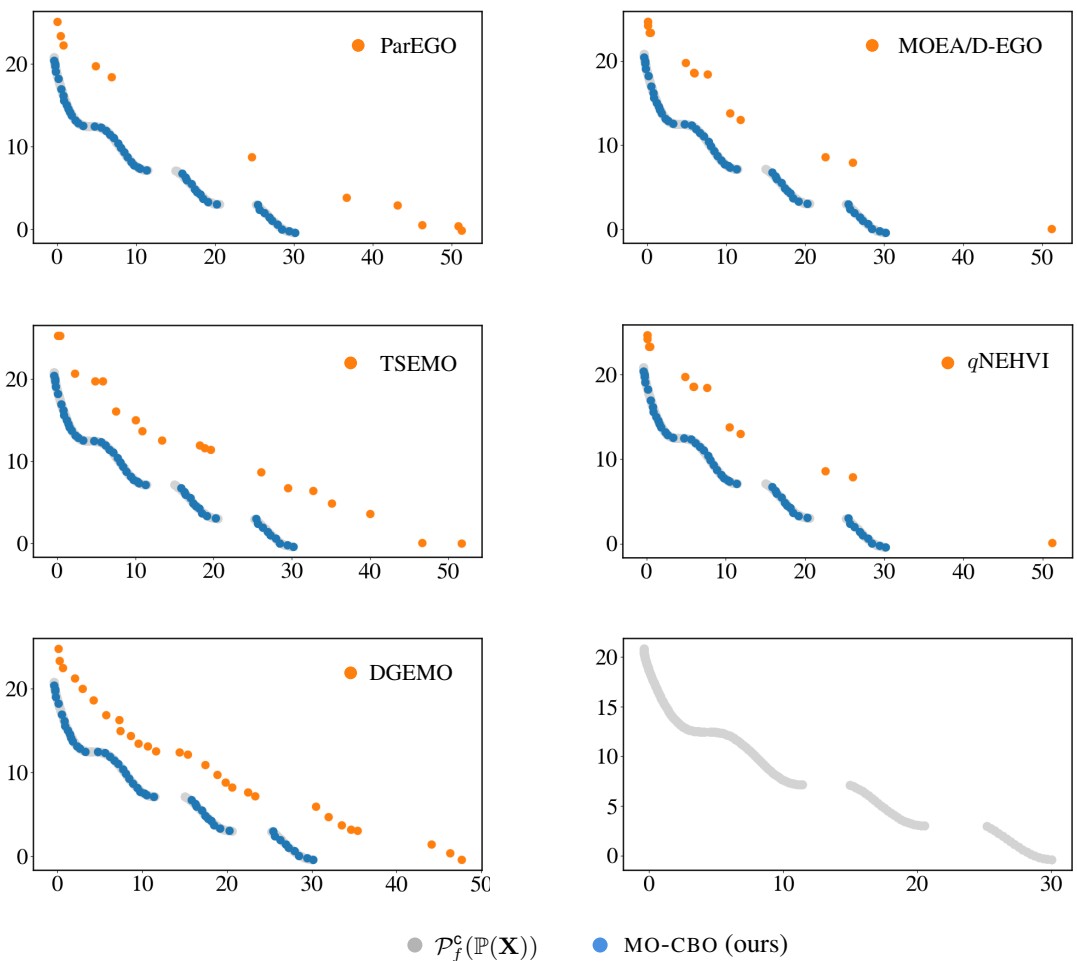

Figure 14. SYNTHETIC-2. Comparison of Pareto front approximations produced by MO-CBO (ours) and various MOBO baselines: ParEGO, MOEA/D-EGO, TSEMO, $q$NEHVI, and DGEMO. The $x$-axis corresponds to objective $Y_1$, and the $y$-axis to $Y_2$.

**HEALTH** We present the experimental results for the MO-CBO problem HEALTH, see Figure 15. We observe that the solutions identified by our MO-CBO algorithm tightly fit the target Pareto front $\mathcal{P}_f^{\mathsf{c}}(\mathbb{P}(\mathbf{X}))$, and strictly dominate those found by the MOBO baselines. All methods were run until a fixed cost budget of 120 was exhausted.

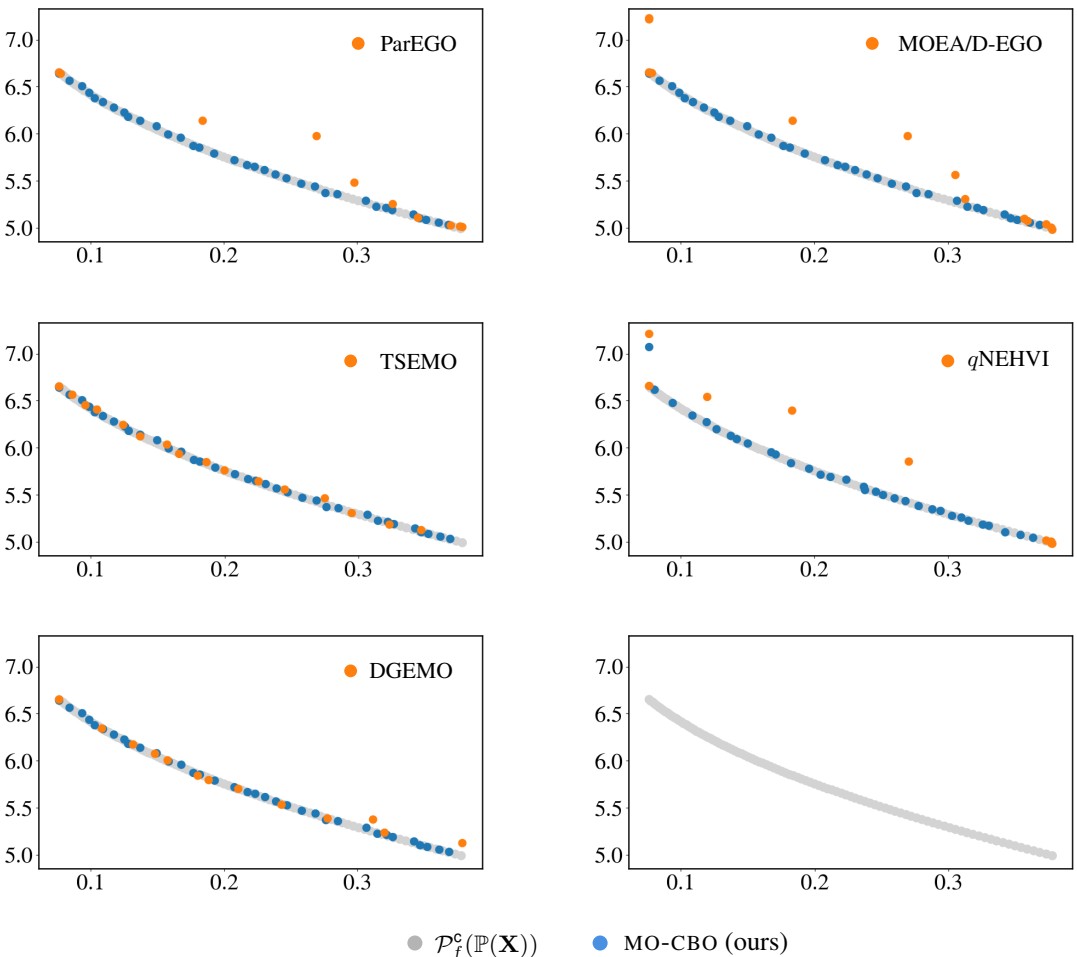

*Figure 15.* HEALTH. Comparison of Pareto front approximations produced by MO-CBO (ours) and various MOBO baselines: ParEGO, MOEA/D-EGO, TSEMO, $q$NEHVI, and DGEMO. The $x$-axis corresponds to objective $Y_1$, and the $y$-axis to $Y_2$.

**CREDIT APPROVAL** We present experimental results for the MO-CBO problem CREDIT APPROVAL in Figure 16. Our MO-CBO algorithm consistently outperforms the MOBO baselines, producing solutions that are both more diverse and better distributed across the target Pareto front. The experiments are executed until a cost budget of 300 is reached.

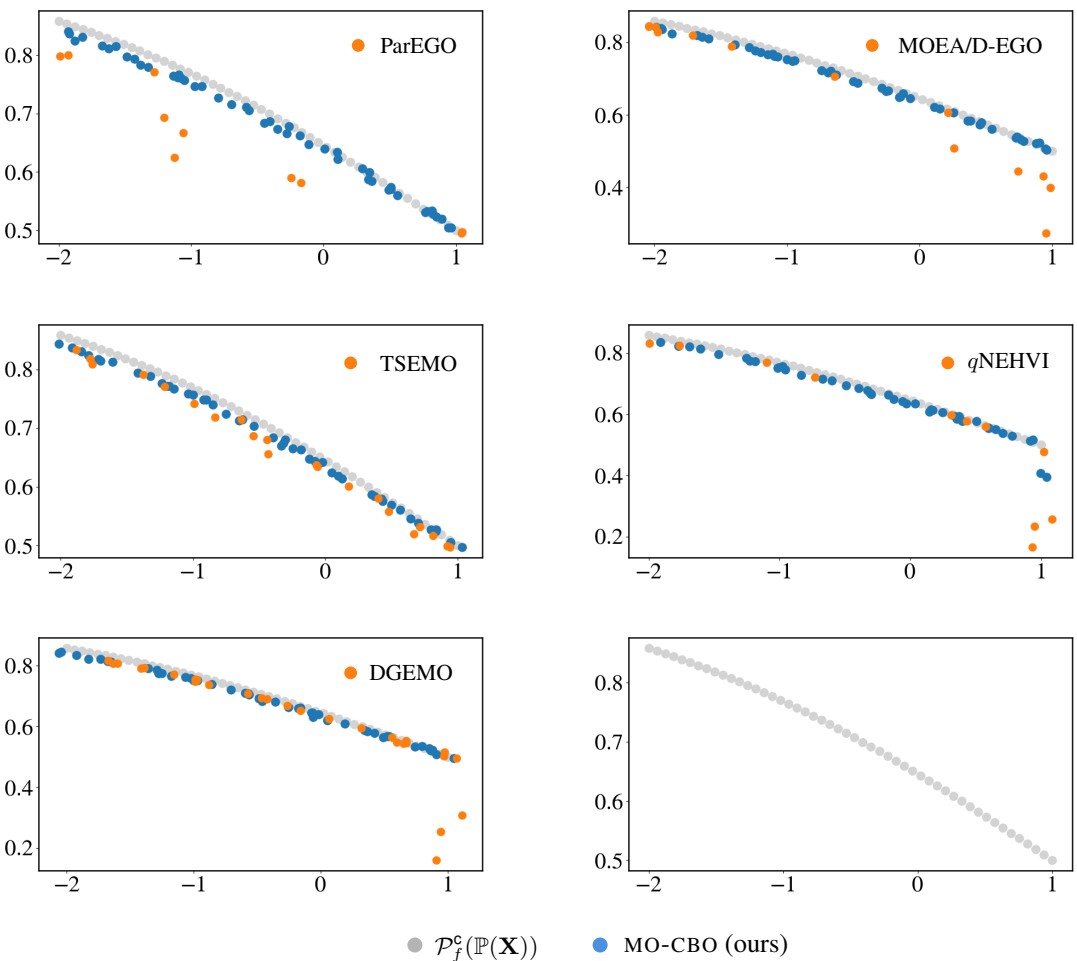

*Figure 16.* CREDIT APPROVAL. Comparison of Pareto front approximations produced by MO-CBO (ours) and various MOBO baselines: ParEGO, MOEA/D-EGO, TSEMO, $q$NEHVI, and DGEMO. The $x$-axis corresponds to objective loan duration, and the $y$-axis to the approval probability.

