# OpenReview forum: "Multi-Objective Causal Bayesian Optimization"
_ICML.cc/2025/Conference — ICML 2025 poster_

### Official Review · Reviewer_8ruG · 2025-02-24

**Overall Recommendation:** 2

**Summary:**

The paper introduces a novel framework, MO-CBO, which integrates causal inference with multi-objective Bayesian optimization, addressing an underexplored research area. The theoretical characterization of Pareto-optimal intervention sets via causal graph topology is a notable contribution. However, several aspects of the paper require improvement.

**Claims And Evidence:**

yes

**Essential References Not Discussed:**

yes

**Experimental Designs Or Analyses:**

yes

**Methods And Evaluation Criteria:**

yes

**Other Comments Or Suggestions:**

None

**Other Strengths And Weaknesses:**

Strength:

Introduces a new class of optimization problems, MO-CBO, expanding causal Bayesian optimization to multi - objective scenarios, filling a research gap in multi-objective optimization considering causal structures.

Weakness:

Although it mentions the potential combination of existing CBO variants with MO-CBO in the future, the paper does not explore this aspect, lacking the full integration of existing research results and limiting the depth and breadth of the research.

**Questions For Authors:**

a.	The paper lacks a thorough discussion of the current state of the field, making the motivation for this work somewhat unclear. A more explicit analysis of existing challenges and the necessity of this approach would strengthen the introduction.
b.	While the theoretical analysis (e.g., Propositions 3.4, 4.10) is rigorous, and the proofs in the appendix enhance the credibility of the results, the assumption of a fully known causal graph is restrictive and may not be practical in real-world scenarios. The paper should discuss how partial knowledge of the causal structure (e.g., unobserved confounders) could impact the algorithm’s performance. Additionally, the surrogate model assumes independent Gaussian processes for each objective, but a discussion on how to handle shared confounders or leverage multi-task learning would improve the methodological robustness.
c.	The proof of section 4.1 derivations should be presented in a more symbolic and structured manner to enhance readability.
d.	While the paper is generally well-written, some sections lack clarity. For example, the construction of the structural causal model (SCM) in Appendix B.2 (Theorem 4.8 proof) is overly technical and difficult to follow.
e.	The baseline comparisons appear to be limited to DGEMO (2020). To convincingly demonstrate the advantages of causal Bayesian optimization, the study should compare against more recent and efficient multi-objective optimization algorithms.
f.	Figure 7 presents experimental results comparing MO-CBO with MOBO on a real-world health application, where MO-CBO demonstrates superior performance. However, additional real-world applications should be explored to further substantiate the practical utility of MO-CBO.

**Relation To Broader Scientific Literature:**

Related to some extent

**Theoretical Claims:**

yes

---

> ### Author Rebuttal · Authors · 2025-03-31
>
> Dear Reviewer 8ruG,
>
> Thank you for your insightful suggestions to improve our work with a more extensive literature review, better defined limitations, and improved experimental evaluation! We hope that our response below will further reinforce your confidence in our work:
>
> # Extensions to MO-CBO
>
> Our contribution is the development of a multi-objective variant of CBO, MO-CBO, for which we theoretically and empirically show superior performance compared to traditional MOBO. We introduce MO-CBO as its own problem formulation and our paper aims to provide a methodology to address this new type of problem. This includes a decomposition of the MO-CBO task along with establishing graph characterizations to identify possibly Pareto-optimal sets to intervene upon. Our empirical validation supports the theoretical claims.
>
> **Combining MO-CBO with existing CBO variants.** There exist many CBO variants, each developed through its own dedicated research effort - examples include dynamic CBO, functional CBO, and constrained CBO. In this sense, our work stands as a distinct research contribution in its own right. We find it more appropriate to explore these integrations in future research.
>
> **Prior knowledge on the causal structures.** Requiring prior knowledge about the causal graph is indeed a notable limitation of causal Bayesian optimization methods, including our own work. To better communicate this, we now added a limitation section (see our response to reviewer easG for more details on the applicability of our method under causal knowledge requirements). We would like to emphasize that investigating how MO-CBO performs under partial knowledge of the causal structure should be the focus of a separate research effort aimed at developing graph-agnostic (MO-)CBO methods (e.g. see Mukherjee et al., (2024)).
>
> **Multi-task GPs.** Multi-task GPs are a possible technique to increase cost-effectiveness of MO-CBO. We had already discussed this at the end of the paper as a limitation of our work since we believe that an extensive study on multi-task GPs for MO-CBO is best dedicated to future work.
>
> # Literature Review
>
> We do agree with you that our review of the current state of the field would benefit from a more thorough analysis of existing literature! Therefore, we included a review on multi-armed bandit problems as well as a more detailed description of existing CBO variants in order to give a better overview of the methods within causal decision-making. Moreover, we discuss the assumption of having fully vs. partial knowledge of the causal graph in the prior works.
>
> # Proofs
>
> **Intuition of proofs.** Thank you for pointing this out! We did restructure some of the proofs to enhance readability - see our response to reviewer fTDx for a more structured proof of Proposition 4.2. However, we still believe that maintaining mathematical rigor requires some level of technical detail. That said, to enhance readability, we have now included a description of the main idea of each proof within 1-2 sentences in the main body of the paper. For instance, for Proposition 3.4 we write: “The rigorous proof of Proposition 3.4 is given in Appendix A, and it exploits the fact that the space of all intervention set-value pairs is the union of the input spaces of each local problem. This allows to match the Pareto optimal intervention set-value pairs with the Pareto-optimal solutions from the local problems where the intervention set is fixed.”
>
> **Proof of Theorem 4.8.**  In Theorem 4.8 we show that any intervention on $\mathbf{X}\_s$ is weakly-dominated by some intervention on  $\textnormal{IB}(\mathcal{G}\_{\overline{\mathbf{X}}\_s}, \mathbf{Y})$. Thus, we did not construct an SCM here. We assume you mean Proposition B.2 or Proposition 4.7? We are happy to answer any remaining questions regarding these proofs!
>
> # Experiments and Baselines
>
> **Additional real-world experiment.** Following your suggestion, we include one more real-world problem, where the SCM describes macro-economic relations based on real-world data (Höllig et al., 2023). Due to space limitations, we would like to point you to our response to reviewer 7sGg for preliminary results.
>
> **MOBO baselines.** We expand our ablation study to include the MOBO algorithms DGEMO, TSEMO, ParEGO, MOEA/D-EGO, qParEGO and qNEHVI. We find that our method performs better than all baselines across all tasks. For more details (including preliminary results) we refer to our response to reviewer 7sGg.
>
> Thanks again for your thoughtful comments - they greatly contribute to improving the clarity of our work! We hope our explanations have addressed your concerns, and we look forward to hearing back.
>
> ## References
> Mukherjee et al. Graph Agnostic Causal Bayesian Optimisation. In NeurIPS 2024 Workshop on Bayesian Decision-making and Uncertainty.
>
> Höllig et al. Semantic meaningfulness: Evaluating counterfactual approaches for real-world plausibility and feasibility, in: xAI, 2023, pp. 636–659.

---

### Official Review · Reviewer_easG · 2025-03-14

**Overall Recommendation:** 3

**Summary:**

The paper considers the problem of multi-objective optimisation with knowledge of the causal graph of the underlying system, where the objectives are the interventional means of a set of target variables. The authors propose a bayesian optimisation solution and prove theoretically that knowledge of the causal graph provides advantages by reducing the search space for optimisation. The authors prove, using synthetic data, that their method is better than the baselines of no knowledge of causality.

**Claims And Evidence:**

I think the claims are supported, with the exception of the applicability of which I talk in more detail in the weaknesses section.

**Essential References Not Discussed:**

Not that I am aware of.

**Experimental Designs Or Analyses:**

The experimental analyses seem reasonable to me. I would not personally call the health problem a real-world problem since, in the end, it is using synthetic functions to evaluate the model. Additionally, I would not say “Weight” and “BMI” are easy variables to treat.

**Methods And Evaluation Criteria:**

I am unsure. I am not aware of the GD and IGD metrics to evaluate multi objective optimisation problems. I think it would be beneficial for the unknowledgeable readers like me to have a very short introduction on the appendix. In addition I would also include the definition of HVI and $\mathcal{H}$ in the appendix.

**Other Comments Or Suggestions:**

- There is a LaTeX compilation error on Figure 2 below  “Solve the local problems”.
- Line 122, second column should be a subset symbol instead of a set membership symbol, I believe.
- Line 124, second column, any reason for using lowercase $x$ instead of  $\bigtimes$ (\bigtimes) (or $\Pi$ (\Pi)) for the product space?
- It seems this is intentional but I find it strange that in line 181 and 202, second column; 273 first column; $\mu_i$ instead of just $\mu$ but then you refer to “for all $i$”.

**Other Strengths And Weaknesses:**

Strengths:
- I think the approach proposed by the authors is sensible and fills some of the gaps of the literature.
- The theoretical analysis is serious and exploits some interesting characteristics of causality.

Weaknesses:
- I find the applicability of the method to be limited. Not in the sense that the method is not –theoretically– general enough, but in the sense that the conditions for the method to be applied are too constrained. When do such multi-objective optimisation problems arise in real world problems where in addition we have knowledge of the causal graph?
- The authors could say that the health problem is one such possibility, but as I argued above, weight and BMI are not easily manipulable. But even beyond that, suppose that the health problem is a real world multiobjective optimisation with known causal graph. Then intervening on the system would render a different causal graph (where the arrows of the intervened variables are deleted) and the optimisation becomes a different problem? Maybe I’m missing something here but it would be interesting to understand this a bit better.

**Questions For Authors:**

Please see weaknesses.

**Relation To Broader Scientific Literature:**

This paper extends the research in two clear directions, one relating to the use of causality in Bayesian optimisation and the other relating to multiobjective Bayesian optimisation. I think the approach that the authors are proposing is sensible with respect to what has been already said in these two areas.

**Theoretical Claims:**

The only proof I read in detail is the proof on the paper. If the rest of the proofs are of the same quality and clarity, I don’t think there should be any problem with those.

---

> ### Author Rebuttal · Authors · 2025-03-31
>
> Dear Reviewer easG,
>
> Thank you for recognizing our contributions and for your insightful feedback pointing out the need to better discuss the applicability of our approach and adding further evaluations.
>
> # Applicability of MO-CBO
> Requiring prior knowledge about the causal graph is indeed a notable limitation of current causal optimization methods and of our work. We acknowledge that this limitation was not highlighted sufficiently in the initial version of our paper. To better communicate this, we now added a limitation section.
>
> Firstly, in many domains causal knowledge is readily available. For example, the effects of drugs are studied in experimental trials in the medical field. Knowledge about the effects, side-effects, and interactions is available and an example of causal relationships that can be used for CBO. Secondly, if causal information is not readily available, there exist methods to discover it, falling  under the field Causal Discovery (Zanga et al., 2023). Here, the corresponding methods aim to identify causal structures from data.
>
> **Health Problem.** We refer to the Health example as a real-world problem since the relations between the variables were derived from real-world analyses (Ferro et al., 2015). We agree that weight and BMI are not easy-to-treat variables. However, they are treatable under high costs. We increased the cost of intervention for the BMI and weight variables in our experiment and added a respective discussion. As a result, the distinction between MOBO and MO-CBO is even clearer, with MO-CBO demonstrating a significantly higher coverage of the causal Pareto front compared to MOBO. The IGD under MO-CBO is 0.02 whereas the IGD under MOBO is 0.05 in an output space ranging from values 0.05 to 0.4 for the Statin medication (thus the difference in IGD is not negligible) .
>
> **Economics Problem.** We include one more real-world problem, where the SCM describes macro-economic relations based on real-world data  (Höllig et al., 2023). Due to space limitations, we would like to point you to our response to reviewer 7sGg for preliminary results.
>
> # Graph Representation of Interventions
> Regarding your question if intervening on the system would render a different causal graph: An intervention on variables $\mathbf{X}\_s \subseteq \mathbf{X}$ involves replacing the structural equations $f\_{X}$ with a constant $x$ for all $X \in \mathbf{X}\_s$, denoted $\text{do}(\mathbf{X}\_s = \mathbf{x}\_s)$. Thus, performing an intervention removes the influence of all variables on $\mathbf{X}\_s$. The graph $\mathcal{G}\_{\overline{\mathbf{X}}\_s}$ represents this intervention and is obtained by removing incoming edges into $\mathbf{X}\_s$. Therefore, $\mathcal{G}\_{\overline{\mathbf{X}}\_s}$ is used to denote the causal structure under intervention and to perform theoretical analyses regarding the effects of interventions. Importantly, the optimization problem itself remains unchanged.
>
> # Performance Metrics
> We added the definitions of GD and IGD from Schulze et al. (2012). They are as follows:
> Let $\mathbf{A}$ be the set of points from an approximated Pareto front and let $\mathbf{Z}$ be the set of points on the true Pareto front.
>
> **GD.** The GD is the average distance from any point $\mathbf{a}\_i \in \mathbf{A}$ to its closest point in the Pareto front $\mathbf{Z}$. Formally,
>              $$\text{GD}(\mathbf{A},\mathbf{Z}) =  \biggl( \ \frac{1}{|\mathbf{A}|} \sum\_{i=1}^{|\mathbf{A}|} d\_i^p \ \biggr)^{1/p},$$
> where $d\_i$ is the Euclidian distance from $\mathbf{a}\_i$ to its nearest point in $\mathbf{Z}$; we set $p=2$ in our experiments.
>
> **IGD.** The IGD measures the average distance from any point $\mathbf{z}\_i \in \mathbf{Z}$ to its closest point in $\mathbf{A}$.
> Formally,
> $$\text{IGD}(\mathbf{A},\mathbf{Z}) = \biggl( \ \frac{1}{|\mathbf{Z}|} \sum\_{i=1}^{|\mathbf{Z}|} \hat{d\_i}^p \ \biggr)^{1/p},$$
> where $\hat{d\_i}$ is the Euclidian distance from $\mathbf{z}\_i$ to its nearest point in $\mathbf{A}$.
>
> The GD evaluates the convergence of the approximated Pareto front to the true front, whereas the IGD measures the diversity of the solutions across the output space.
>
> We also implemented your remaining suggestions and included the definition of the hypervolume in the Appendix. We hope our explanations fully address your concerns, and we look forward to your response!
>
> ## References
> Zanga et al. 2022. A survey on causal discovery: Theory and practice.Int. J. of Approximate Reasoning, 151 (2022), 101–129.
>
> Schutze et al. (2012). Using the averaged hausdorff distance as a performance measure in evolutionary multiobjective optimization. IEEE T. on Evolutionary Computation, 16(4), 504–522.
>
> Ferro et al. (2015). Use of statins and serum levels of prostate specific antigen. Acta Urológica Portuguesa, 32.
>
> Höllig et al. Semantic meaningfulness: Evaluating counterfactual approaches for real-world plausibility and feasibility, in: xAI, 2023, pp. 636–659.

---

### Official Review · Reviewer_7sGg · 2025-03-14

**Overall Recommendation:** 2

**Summary:**

- Decision-making outcomes depend on causal relationships and evaluating them is costly.
- Causal Bayesian optimization uses these relationships to find optimal interventions efficiently.
- Multi-objective causal Bayesian optimization (MO-CBO) extends causal Bayesian optimization to identify Pareto-optimal interventions.
- MO-CBO can decompose into several traditional multi-objective tasks and balance exploration across them.
- MO-CBO is validated on synthetic and real data, outperforming non-causal methods when causal information is available.

**Claims And Evidence:**

- This research tackles an intriguing issue in causal Bayesian optimization.
- The theoretical results strongly back the claims of this study, but I have some concerns regarding the experiments.

**Essential References Not Discussed:**

- I don't think there are specific references not discussed.

**Experimental Designs Or Analyses:**

- All evaluations at intervention 0 should be identical across MO-CBO and MOBO, if the same random seeds are given for both methods.
- I think the authors can add more baselines such as qNEHVI and ParEGO.
- Why are the results of MO-CBO and MOBO similar in Synthetic 1?  Both just work well in this problem?  If there is a particular message of this problem, it might be able to be removed.
- The results of the Health problem are also similar across two methods.  Why do both methods yield similar results?

**Methods And Evaluation Criteria:**

- Can you provide mathematical definitions of GD and IGD?
- What are PSA and Stain in the Health problem?
- How can we choose $m$ for the number of independent Gaussian processes?

**Other Comments Or Suggestions:**

- Are there additional real-world problems to consider? Including more could enhance the presentation of this work.
- In Figure 2, there is Sec. ??.  It should be fixed.

**Other Strengths And Weaknesses:**

Please see above.

**Questions For Authors:**

- As $m$ becomes large, the computational cost of building a surrogate model increases.
- Could you provide the run-time results for the proposed method?
- How can the causal Pareto front be identified? Is there a difference between determining the standard Pareto front and the causal Pareto front?

**Relation To Broader Scientific Literature:**

- This work aligns with causal Bayesian optimization and broader scientific literature on Bayesian optimization.

**Theoretical Claims:**

- The authors can provide the proof sketches of the propositions and theorems.

---

> ### Author Rebuttal · Authors · 2025-03-31
>
> Dear Reviewer 7sGg,
>
> Thank you for your thoughtful feedback and for recognizing our contributions! We have incorporated all of your suggestions to enhance readability, include additional baselines, and provide a clearer formulation of the causal Pareto front for the community.
>
> # Baselines
> Thank you for your suggested improvements to our experimental analysis! We now include the MOBO algorithms DGEMO, TSEMO, ParEGO, MOEA/D-EGO, qParEGO, and qNEHVI as baselines. Additionally, we added another real-world problem, where the SCM describes macro-economic relations and is based on real-world data (Höllig et al., 2023). The average GDs are (qParEGO and qNEHVI are still being implemented):
>
> |       |  DGEMO  |  TSEMO  |  ParEGO  |  MOEA/D-EGO |  MO-CBO (ours) |
> |----|:----:|:----:|:----:|:----:|:----:|
> | Synthetic-1 | **0.14** | 0.16 | 0.30 | 0.38 |**0.14**|
> |  Synthetic-2 | 7.55 | 7.78 | 12.50 | 11.95 |**1.85**|
> | Health | 0.07 | 0.07 | 0.09 | 0.20 |**0.06**|
> | Economics |  4.66  | 4.93 | 5.56 | 6.41 |**1.22** |
>
> Analyzing both GD and IGD, we find that our method performs better than all baselines across all tasks.
>
>
> # Experiments
> **IGD and GD values at zero interventions.** Thank you for pointing this out - this is a x-ticks label error. Our algorithm is initialized with prior data that is used to compute a first estimate of the causal Pareto front before starting the optimization loop. We now label the start of the plots with len(prior data) = 5 to resolve this unclarity.
>
> **Similarity between MOBO and MO-CBO.** It is theoretically grounded from Proposition 4.4 that both methods will converge to the same results when no $Y\_i$ is confounded with its ancestors via unobserved confounders, which is indeed the case for both Synthetic-1 and the Health example. In these scenarios, MOBO intervenes on all treatment variables $\mathbf{X}$, while MO-CBO intervenes only on the parents of $\mathbf{Y}$. Note that intervening on $\mathbf{X}$ yields the same results as restricting the intervention to the parents of $\mathbf{Y}$. As shown in Fig. 4, MO-CBO identifies a greater number of solutions by avoiding unnecessary interventions on elements outside the minimal intervention set defined by the parents of $\mathbf{Y}$. This approach reduces costs while achieving broader coverage of the causal Pareto front. However, when some $Y\_i$ is confounded with its ancestors via unobserved confounders, our method can establish superior results than traditional MOBO approaches as demonstrated in Fig. 6 for Synthetic-2.
>
> Moreover, we added the definitions of our metrics as well as the runtime results to the Appendix. All methods, including ours, have an average runtime per iteration between 0.3 and 14 seconds. The fastest method is TSEMO with 0.2 - 4 seconds per iteration (depending on the problem) while ours ranges between 3 and 6 seconds.
>
> # General Discussion
> **Number of independent Gaussian Processes.** The number of independent Gaussian processes (GP) $m$ is equal to the number of target variables, we describe this in the preliminaries of MO-CBO in Section 2. We train the GPs independently from each other, resulting in a linear increase of computational cost with $m$.
>
> **Proof Sketches.** Thank you for this suggestion - we think this greatly improves the overall understanding of our methodology! We have now included a description of the main idea of each proof within 1-2 sentences in the main body of the paper. For instance, for Proposition 3.4 we write: “The rigorous proof of Proposition 3.4 is given in Appendix A, and it exploits the fact that the space of all intervention set-value pairs is the union of the input spaces of each local problem. This allows to match the Pareto optimal intervention set-value pairs with the Pareto-optimal solutions from the local problems where the intervention set is fixed.”
>
> **Causal Pareto front.** In MO-CBO, the input space comprises all possible intervention set-value pairs. Based on proposition 3.4, the causal Pareto front is constructed by identifying the Pareto-optimal points across the solutions of local problems, each corresponding to a fixed intervention set. However, since *Pareto front* is a well-established concept in MOBO, we acknowledge that the phrase "causal Pareto front" can cause confusion. To clarify, the causal Pareto front simply represents the best attainable trade-offs *within in a causal system*, considering intervention set-value pairs as inputs. Based on your comment, we propose to rename the term to *Pareto front in causal systems*.
>
> We have addressed all of your remaining comments as well as added the clarification on PSA (prostate specific antigen) and Statin (a type of medication) to the experiments section. Thanks again for your comments to enhance the clarity of our work! We look forward to your response.
>
> ## Reference
> Höllig et al. Semantic meaningfulness: Evaluating counterfactual approaches for real-world plausibility and feasibility, in: xAI, 2023, pp. 636–659.

---

### Official Review · Reviewer_fTDx · 2025-03-15

**Overall Recommendation:** 4

**Summary:**

This paper presents a unification of Multi-objective Bayesian Optimisation and Causal Bayesian Optimisation to causal settings in which dependence between variables is mediated by a causal graph, maintaining the same assumptions of a known causal graph but not the structural equations or exogenous distribution as CBO. This results in a procedure for performing Multi-objective Causal Bayesian Optimisation in such settings. The authors develop theory that allows them to reduce the search space down from the power set over manipulative variables to a smaller set of so-called "possibly Pareto-optimal minimal intervention sets", which motivates an algorithm that assesses only possibly Pareto-optimal minimal intervention sets to solve MOCBO problems. The performance of their method is validated in three experiments in which a non-causal Multi-objective Bayesian Optimisation procedure acts as a baseline.

**Claims And Evidence:**

The theory and experiments support the claims made by the authors.

**Essential References Not Discussed:**

None that come to mind.

**Experimental Designs Or Analyses:**

As discussed above, I think the experiments are selected well to highlight the value of this method over non-causal counterparts.

**Methods And Evaluation Criteria:**

I think the experiments to evaluate their method are well-designed and demonstrate the value of the proposed method. They demonstrate this by considering settings in which there are and are not hidden confounders to highlight the overlap in and differences between MO-CBO and MOBO.

**Other Comments Or Suggestions:**

Page 4, just before Proposition 4.4, there is a typo: you write "...which we **proof** in Appendix B.2..." but you want "...which we **prove** in Appendix B.2..."

**Other Strengths And Weaknesses:**

Further strengths are: notation is clear and defined well up front (I found it necessary to refer back to earlier parts of the work to look up notation, but could always find definitions clearly stated earlier); and I appreciated the two **Examples** the authors placed in Section 4 to help with intuition and clarity.

**Questions For Authors:**

Please see **Theoretical claims** above.

**Relation To Broader Scientific Literature:**

The paper relates well to relevant prior literature in both causal decision-making and multi-objective Bayesian optimisation (MOBO). In particular, the method naturally extends upon the DGEMO algorithm for non-causal MOBO, and the benefits of this innovation are clearly demonstrated in the experiments.

**Theoretical Claims:**

I went through all proofs. The only thing that struck me as potentially problematic (but more likely a misunderstanding on my part, hence I would appreciate clarification from the authors please) was the proof for Proposition 4.2. My concern is that the proof appears limited in generality under the assumption that all nodes are associated with their own fair binary exogenous variable, and that the structural equations are just sums over the values of the parent nodes. Why is it ok to consider this case only to prove what looks like a very general claim in Proposition 4.2?

Also, is Proposition 4.2 a new result? I am not sure that I have an especially well-calibrated sense of what is significant in this field, but this result seems like it deserves more attention than the authors are currently giving it?

---

> ### Author Rebuttal · Authors · 2025-03-31
>
> Dear Reviewer fTDx,
>
> We greatly appreciate your detailed feedback and your recognition of our contributions! We are happy to address your remaining questions below:
>
> # Proposition 4.2
> Proposition 4.2 is based on Proposition 1 of Lee & Bareinboim (2018), which formalizes the concept of minimal intervention sets for graphs with a single target variable $Y$. We extend this result to causal graphs with multiple target variables $Y\_1,…,Y\_m$.
>
> Your question refers to the “if” direction of the proof. Let us first state the definition of minimal intervention sets along with the proposition.
>
> **Definition 4.1** (Minimal intervention set)**.** A set $\mathbf{X}\_s \in \mathbb{P}(\mathbf{X})$ is called a *minimal intervention set* if there exists no subset $\mathbf{X}\_{s}' \subset \mathbf{X}\_s$ such that for all $\mathbf{x}\_s \in \mathcal{D}(\mathbf{X}\_s)$ it holds $\mu(\mathbf{X\_s},\mathbf{x}\_s) = \mu(\mathbf{X}\_{s}',\mathbf{x}\_s[\mathbf{X}\_{s}'])$, $1 \leq i \leq m$, for every SCM conforming to $\mathcal{G}$.
>
> **Proposition 4.2** Given a causal graph $\mathcal{G}$, $\mathbf{X}\_s \in \mathbb{P}(\mathbf{X})$ is a minimal intervention set if and only if it holds $\mathbf{X}\_s \subseteq \textnormal{an}(\mathbf{Y})\_{\mathcal{G}\_{\overline{\mathbf{X}}\_s}}$.
>
> # Proof Strategy
>
> The "if" direction of the proposition states: $\mathbf{X}\_s \subseteq \textnormal{an}(\mathbf{Y})\_{\mathcal{G}\_{\overline{\mathbf{X}}\_s}} \implies \mathbf{X}\_s$ is a minimal intervention set. To prove this, we use contraposition, meaning we will show the equivalent statement: $\mathbf{X}\_s$ is not a a minimal intervention set $\implies$ $\mathbf{X}\_s \not\subseteq \textnormal{an}(\mathbf{Y})\_{\mathcal{G}\_{\overline{\mathbf{X}}\_s}}$. We show this contrapositive via contradiction: Suppose $\mathbf{X}\_s$ is not a minimal intervention set, but assume for contradition $\mathbf{X}\_s \subseteq \textnormal{an}(\mathbf{Y})\_{\mathcal{G}\_{\overline{\mathbf{X}}\_s}}$. By definition 4.1, there exists a subset $\mathbf{X}\_{s}' \subset \mathbf{X}\_s$ such that **for all SCMs conforming to** $\mathcal{G}$, intervening on $\mathbf{X}\_{s}'$ yields the same outcome as intervening on $\mathbf{X}\_s$, i.e., $\mu(\mathbf{X}\_s,\mathbf{x}\_s) = \mu(\mathbf{X}\_{s}',\mathbf{x}\_s[\mathbf{X}\_{s}'])$, where $\mathbf{x}\_s [\mathbf{X}\_{s}']$ are the values of $\mathbf{x}\_s$ corresponding to $\mathbf{X}\_s \cap \mathbf{X}\_{s}'$. We construct a specific SCM that conforms to $\mathcal{G}$, meaning that we assign structural equations between the variables within $\mathcal{G}$. Then, we leverage these structural equations along with the assumption $\mathbf{X}\_s \subseteq \textnormal{an}(\mathbf{Y})\_{\mathcal{G}\_{\overline{\mathbf{X}}\_s}}$ to establish a contradiction, which is given by $\mu(\mathbf{X\_s},\mathbf{x}\_s) > \mu(\mathbf{X}\_{s}',\mathbf{x}\_s[\mathbf{X}\_{s}'])$, thereby breaking the equality. This invalidates our initial assumption, yielding $\mathbf{X}\_s \not\subseteq \textnormal{an}(\mathbf{Y})\_{\mathcal{G}\_{\overline{\mathbf{X}}\_s}}$. We have thus proven the contrapositive statement.
>
> **Generality.** Since $\mathbf{X}_s \not\subseteq \textnormal{an}(\mathbf{Y})\_{\mathcal{G}\_{\overline{\mathbf{X}}\_s}}$ is a graph-topological property (i.e. it does not associate to any specific SCM), the above strategy completes the proof. Different SCMs cannot yield different graph-topological properties. The constructed SCM serves only as a tool to establish the desired result. We recognize that this aspect may not have been fully articulated in the original proof and have now revised it to improve clarity.  While our proof required some technical modifications to the one by Lee & Bareinboim (2018), the core idea remains unchanged.
>
> Thank you for your time and extensive review of all our proofs. We believe your questions and remarks greatly improve the understanding of our derivations for future readers! We hope this clarification fully addresses your concerns, and we look forward to your response.
>
> ## References
>
>  Lee, S. and Bareinboim, E. Structural causal bandits: Where to intervene? In Advances in Neural Information Processing Systems, 2018.

---

> > ### Comment · Reviewer_fTDx · 2025-04-02
> >
> > Thanks for your response and for clarifying – indeed I think I failed to retain the "for all SCMs conforming to $\mathcal{G}$" while reading through the proof. Nice work, and I'm happy to maintain my score.

---

> > > ### Author Response · Authors · 2025-04-02
> > >
> > > Great, thank you! Happy to have sorted it out.

---

### Decision · Program_Chairs · 2025-05-01

**Decision:**

Accept (poster)

**Comment:**

This manuscript considers the problem of multi-objective causal Bayesian optimization that operates by reducing the problem to solving a series of traditional multi-objective optimization problems, then solving these jointly while balancing exploration among this set. The authors perform a series of experiments on synthetic and real data demonstrating the performance of this approach.

The initial response to this work from the reviewers was somewhat mixed in terms of their scores. However, the reviewers agree that the considered setting is of interest to the ICML audience, and most reviewers agree that the paper is well written and the experiments well designed. The two "weak reject" reviews only point out minor weaknesses that I believe were adequately addressed by the author responses (and one of these two reviews actually doesn't really point out any weaknesses at all).

That said, no champion for the paper emerged during the discussion period, even from the reviewer with the highest rating. Thus I am only comfortable recommending weak acceptance here, supported by the apparent soundness of the approach and an assessment that it would be an interesting addition to the program.